# Pseudo-Karst Silicification Related to Late Ni Reworking in New Caledonia

Michel Cathelineau [1,*] , Marie-Christine Boiron [1] , Jean-Louis Grimaud [2] , Sylvain Favier [1] , Yoram Teitler [1] and Fabrice Golfier [1]

1    Université de Lorraine, CNRS, GeoRessources, F-54000 Nancy, France
2    PSL University/MINES Paris/Centre de Géosciences, 35 rue St Honoré, 77305 Fontainebleau Cedex, France
*    Correspondence: michel.cathelineau@univ-lorraine.fr

**Abstract:** Silicification in New Caledonian pseudo-karsts developed on peridotite was assessed using $\delta^{18}O$ and $\delta^{30}Si$ pairs on quartz cements. The objective was to document the chronology of pseudo-karst development and cementation relative to geomorphic evolution. The latter began at the end of the Eocene with the supergene alteration of peridotites and the subsequent formation of extended lateritic weathering profiles. Neogene uplift favoured the dismantling of these early lateritic profiles and valley deepening. The river incision resulted in (i) the stepping of a series of lateritic paleo-landforms and (ii) the development of a pseudo-karst system with subvertical dissolution pipes preferentially along pre-existing serpentine faults. The local collapse of the pipes formed breccias, which were then cemented by white quartz and Ni-rich talc-like (pimelite). The $\delta^{30}Si$ of quartz, ranging between −5‰ and −7‰, are typical of silcretes and close to the minimum values recorded worldwide. The estimated $\delta^{18}O$ of −6 to −12‰ for the fluids are lower than those of tropical rainfall typical of present-day and Eocene–Oligocene climates. Evaporation during drier climatic episodes is the main driving force for quartz and pimelite precipitation. The silicification presents similarities with silcretes from Australia, which are considered predominantly middle Miocene in age.

**Keywords:** silcrete; pseudo-karst; laterite; nickel; oxygen and silicon isotopes; weathering





## 1. Introduction

Karst is a geomorphological structure resulting from the hydro-chemical and hydraulic erosion of all soluble rocks. The main karsts are those developed in limestones, which are particularly soluble when subjected to dissolved $CO_2$-rich water. Other rock types can be affected by dissolution leading to geomorphological structures similar to those developed in carbonate rocks; in particular, other sedimentary formations such as evaporites (gypsum or salt layers).

A lesser-known type is dissolution conduits that form in ultramafic rocks such as peridotites, which are soluble when subjected to a warm, rainy climate for a prolonged time. They have been described as "karsts" by several authors, including [1–3] for New Caledonia, but also in Greece [4,5], Cuba [6], Papua New Guinea and Indonesia [7,8] or "pseudo-karsts" [9,10]. In this paper, the term "pseudo-karst" will be used.

When peridotites are submitted to a hot and humid climate, the complete olivine dissolution yields a residual soil, the laterite. The latter is formed essentially of goethite and hematite due to the very low solubility of $Fe^{3+}$ iron oxides and hydroxides under oxidising conditions [11–14]. In the lower part of the lateritic profile, Mg–Ni-silicates form at the boundary with the bedrock in the so-called saprolite horizon [12,14]. The homogeneous lowering of the weathering front and the generation of a weathering horizon of several tens of meters thick implies a duration of at least 10 Ma [15,16].

In New Caledonia, supergene alteration of peridotites started shortly after ophiolite obduction, which occurred between Late Eocene and Early Oligocene [17,18]. This led

to vast expanses of residual laterite soils. Paleomagnetic ages on ferruginous duricrusts suggest that the paleo-weathering is at least as old as 25 Ma [19]. The uplift phases that followed the development of lateritic soils carried the laterites to higher elevations, which triggered erosion after the Oligocene [20]. The relief erosion led to significant loss of the ophiolite, more than two-thirds of the ophiolite surface, and the deepening of the valleys.

Geomorphological changes linked to periods of uplift and erosion can profoundly affect the hydrological regimes and modify the geometry of dissolution patterns. The formation of the pseudo-karst systems was favoured by denudation and valley incision [3,9,21]. This consisted of the complete olivine dissolution along connected fractures and faults, yielding to forming dissolution pipes [9], similar to those defined in other lithological contexts [22]. Locally, the gravity collapse of the pipes caused the formation of dolines at the surface [3,9] and breccia pipes, which were partially cemented locally by white quartz and then by Ni-silicates of a dark green colour. The breccia cement appears, therefore, to be an essential witness of syn- to post-uplift phases, and its analysis may provide keys to the genesis of the preferential Ni enrichments.

Several types of silicification, sometimes so-called silcretes by previous works [2,20], were already identified and linked to the landscape's evolution or the well-known chronology of Ni–ore formation [23–25]. Despite their common occurrence, the mechanisms and timing of silicification in surficial processes remain challenging to constrain [26–28].

Several silicified pipes have been observed in the Koniambo massif, New Caledonia (Figure 1a), mainly from fresh cuts made during open-pit mining operations and digging access tracks to the top of the plateau. These cuts provided a good vertical cross section from bedrock to saprock, with the opportunity to observe and sample breccias preserved from more recent supergene alteration and oxidation. This study aimed to combine the petrological information on the chronology of mineral assemblages with the geomorphic evolution. The conditions of silicification and precipitation of Ni-silicates were derived from a textural study by SEM and cathodoluminescence on breccia quartz cement, combined with a geochemical characterisation (trace elements, $\delta\ ^{30}$Si, $\delta\ ^{18}$O). Field observations were used to constrain the chronology of erosion surfaces with valley incision and to interpret the pseudo-karst system and its cementation in the geomorphic evolution. The resulting conceptual model aimed to unravel the relative timing and geometry of Ni transfer episodes, which is still an open question for future mining prospection.

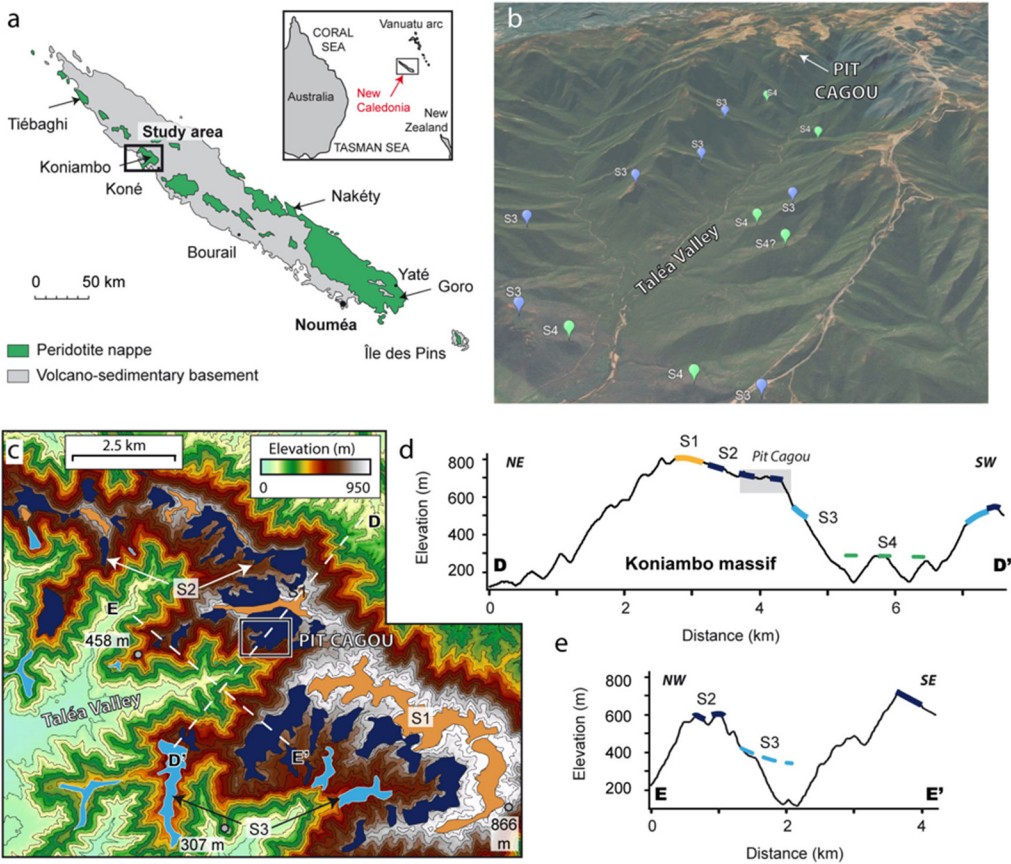

**Figure 1.** (**a**) Location of New Caledonia and Koniambo. (**b**) Aerial view with the indication of paleosurfaces (S1 to S4). (**c**) Location of the study site (open pit Cagou from KNS mining company) in the Koniambo massif. (**d**,**e**) Cross sections are in (**c**) (D-D′ and E-E′), with the indication of S1, S2, S3 paleosurface (in yellow, dark blue and light blue respectively) and the potential location of S4 (green dashed line).

## 2. Geological Setting of the Koniambo Massif

In the Koniambo massif, the Peridotite Nappe is exposed from its base, near sea level, up to ~800 m elevation (Figure 1b,c). The massif essentially consists of harzburgites with interlayers of dunites. In the higher part of the massif, peridotites are moderately serpentinised, contrary to the sole of the nappe, which is highly deformed and serpentinised [29]. At the Koniambo site, numerous serpentine fractures and faults related to syn- to slightly post-obduction stages were reactivated during compressional and extensional phases [29]. These same fractures play an essential role in the drainage of meteoric fluids responsible for the mineralisation of nickel or the current degradation of the old alteration profiles and the formation of the still-active pseudo-karst system [30,31].

Above ~400–600 m in the massif, the nappe is capped by a highly dissected and partly reworked lateritic profile [32]. This is composed of two lateritic remnant surfaces (referred to as S1 and S2), similar to the Tiébaghi plateau further north, as already suggested by Chevillotte (2005) [33].

A succession of ores has been described by Cathelineau et al [23,24] at the Koniambo mine site: (i) Type 1 ore [24] consists of crack-sealed veins with a succession of fillings comprising Mg–Ni talc, followed by red microcrystalline quartz full of microinclusions of iron oxide. These fillings are related to tectonic events that allowed low temperature (50–70 °C) reduced fluids to circulate and mix with oxidising waters, probably at a greater depth than today, (ii) Type 2 ore [23] formed later, closer to the surface, by evaporation in joints developed in metre-sized boulders and consists of films with concentric zones of pimelite (Ni-rich talc) at the periphery and Mg-talc-like in the centre. The ores mined

today (ore Type 3) result from the deepening of the lateritic dissolution front, which affects all previous mineral assemblages and redistributes the nickel into a complex fine-grained talc, nontronite and goethite mineral assemblage. This saprolitic horizon forms a metric to decametric fine-grained layer enclosing preserved boulders and lies between the bedrock and the yellow lateritic horizon where goethite predominates.

## 3. Materials and Methods

Most observations and sampling were made in the Koniambo massif, located along the coast in the northwest part of New Caledonia, during several field campaigns from 2011 to 2021. Samples were taken in the Cagou pit, a location studied in detail for ore Types 1 and 2 and their structural context [29].

### 3.1. Petrographic, Cathodoluminescence and Micro-XRF Images

A Scanning Electron Microscopy (SEM) JEOL J7600F field-effect coupled with SDD type electron dispersive spectrometer, wave wavelength dispersive spectrometer (Oxford Instruments, Abingdon-on-Thames, Oxfordshire, UK) and cathodoluminescence (CITL Cold Cathodoluminescence device Model MK5-1) were used to document mineral assemblages.

Micro-XRF mapping was carried out using the Bruker-Nano M4 Tornado instrument. This system has an Rh X-ray tube with a Be side window and polycapillary optics giving an X-ray beam with a diameter of 25–30 μm on the sample. The X-ray tube was operated at 50 kV and 200 μA and a 2 kPa vacuum. X-rays were detected by a 30 mm$^2$ xflash®SDD with an energy resolution of <135 eV at 250,000 cps. Main elements such as Mg, Mn, Fe, Ni, Co, Cr, and Si were mapped and composite chemical images were generated.

High-Resolution Transmission Electron Microscopy (HRTEM), energy dispersive spectra and electron diffraction patterns were performed on representative samples of pimelite to observe, at a nano-metric scale, the Ni-talc-like texture and particles and obtain an elemental composition. A CM20-Philips instrument with a Si-Li detector operating at 200 kV and 10 eV was used at SCMEM (GeoRessources, Vandœuvre-lès-Nancy, France).

### 3.2. LA-ICPMS Analyses of Trace Elements in Quartz

Trace element abundances in quartz were analysed by a laser ablation inductively-coupled plasma mass spectrometry (LA-ICPMS) at GeoRessources laboratory, University of Lorraine (Vandœuvre-lès-Nancy, France). Analyses were performed using an Agilent 7500c quadrupole spectrometer interfaced with a GeoLas Pro 193 nm ArF excimer laser ablation system (Lambda Physik, Göttingen, Germany). Operating conditions are a 5 Hz repetition rate, a ~10 J/cm$^2$ fluence, and a beam size ranging from 32 to 60 μm. Helium was used as the carrier gas (0.8 L/min) and mixed with Ar before introducing it in the plasma (1.5 L/min). Analysed elements were the following: $^{27}$Al, $^{29}$Si, $^{45}$Sc, $^{49}$Ti, $^{51}$V, $^{53}$Cr, $^{55}$Mn, $^{57}$Fe, $^{59}$Co, $^{60}$Ni, $^{63}$Cu, $^{66}$Zn, $^{69}$Ga, $^{74}$Ge. Acquisition times for background and ablation signals were 40 and 50 s, respectively, allowing the measurement of duplicate spots per analysis. The NIST 610 reference material was used for external and $^{29}$Si for internal standardisation. The NIST SRM 612 glass was employed as a secondary standard and yielded trace element abundances in agreement with the reference values. Data reduction was carried out following the standard methods from Longerich et al. (1996) [34]. The accuracy was around 10% depending on the element.

### 3.3. Oxygen and Silicon Isotopes

The oxygen and silicon isotopic composition of the quartz samples was determined using the CAMECA IMS 1270 ion microprobe at CRPG (Nancy, France) using classical procedures previously described by Rollion-Bard et al. (2007), Robert and Chaussidon (2006) and Marin et al. (2010) [35–37]. Oxygen and silicon isotopic compositions are

reported here as per mil deviations from SMOW standard (Standard Mean Ocean Water) and NBS 28, respectively, using the conventional notation:

$$\delta^{18}O = [(^{18}O/^{16}Osample)/(^{18}O/^{16}O\ SMOW) - 1] \times 1000.$$

$$\delta^{30}Si = [(^{30}Si/^{28}Sisample)/(^{30}Si/^{28}Si\ NBS28) - 1] \times 1000.$$

The sample was analysed with a primary ion beam diameter of 20 μm on standard polished sections coated with gold for oxygen and silicon isotope measurements. The sample was sputtered with a 20 to 30 μm diameter $Cs^+$ primary beam of ~8–10 nA for oxygen measurements and ~20–25 nA for silicon measurements and 10 kV acceleration voltage. Secondary ions were accelerated at 10 kV and detected in multicollection mode. The mass resolving power was set at ≈4000, and the $H_2O$- interference on $^{18}O$-being resolved around 1600. The total analytical time was 4 min, including a presputtering (60 sec), the centring of the magnetic field and analyses (500 sec) for oxygen and silicon measurements. The external reproducibility of the $\delta^{18}O$ measurement was determined using the quartz so-called "Bresil" ($\delta^{18}O$ SMOW= 9.6‰) and was ± 0.20‰ (2σ). The external reproducibility of $\delta^{30}Si$ measurements, determined using the quartz standards NBS28 ($\delta^{30}SiNBS28$= −0.16 ± 0.18), was ± 0.3‰ (2σ).

*3.4. Geochemical Modelling*

The simulations were run on phreeqC code [38,39], with the llnl.dat database edited to account for the Ni-phyllosilicates [16], using data on rainwater from [2]. The Geochemists' Workbench software [40] made the activity diagrams using the thermodynamic data from the edited llnl.dat database.

## 4. Results

*4.1. Field Observations*

Field observations and considering available topographic maps allowed us to determine the relationships among paleosurfaces, S1 to S4, along two perpendicular profiles noted D-D′ and E-E′. S1 and S2 paleosurfaces form two distinctive levels on the plateau, observing the D-D′ topographic profile (Figure 1d). The S1 surface appears as a gently rolling convex hill, while the S2 surface is concave up. The plateau is gently dipping towards the southwest. S1 and S2 are thus in topographic continuity, suggesting a connection between their weathering profiles. Surface 3 (S3) is developed on the southwest foothills of the massif, particularly on the gently westerly dipping low-elevation (120–240 m) Kaféaté plateau along the coast. This observation confirms preliminary conclusions from [41,42]. In the Kaféaté plateau, new field observations indicate a slope break associated with a slight slope change with evidence of reworking of the S3 ferricrete, which is found in the form of boulders in the S4 ferricrete. Hence, this indicates that the S4 surface incised the S3 surface in this area. In the Koniambo massif, relics of the S3 are found at higher elevations (300–600 m), marking the headwaters of erosional glacis developed during the formation of S3 (Figure 1d,e). The silicified rocks and the white quartz from pseudo-karst have similar features to quartz from silcretes on a textural and isotopic basis, as shown later. The term "silcrete" was mentioned by Chardon and Chevillotte (2006) and Chevillotte et al. (2006) [20,42] as a typical weathering product associated with the S4 surface based on the Nepoui cross section on a textural basis. The silcretes are described as silicified conglomerates by Chevillotte et al. (2006) [42] However, it is rather hard to determine if they are similar to the silcrete described in this study, as petrographic descriptions, particularly of the cement, are lacking in their work.

Topographic analysis of the Taléa valley to the southwest of Pit Cagou shows relics of relatively flat surfaces of reduced extent (Figure 1e). Field reconnaissance allowed identifying partly eroded weathering profiles associated with these remnants at lower elevations than S1 and S2 surfaces (Figure 1).

## 4.2. Dissolution Pipes

Near the top of the plateau, in several places, large cavities and open channels are observed in the peridotite. They are best revealed along the main access road to the high plateau (Figure 2). They are located along fractures, mostly filled with serpentine, and follow the inherited fracture network. They are several decimetres to a few metres in aperture and extend over tens of meters in their outcropping part. The bulk vertical extension is inferred from a hydrological study and is about several hundred meters in Koniambo [43]. Only iron hydroxides (goethite) are observed along fracture walls. The protolith is similar on both edges of the fractures and dissolution features are identical to the boulders from the bedrock-saprolite interface.

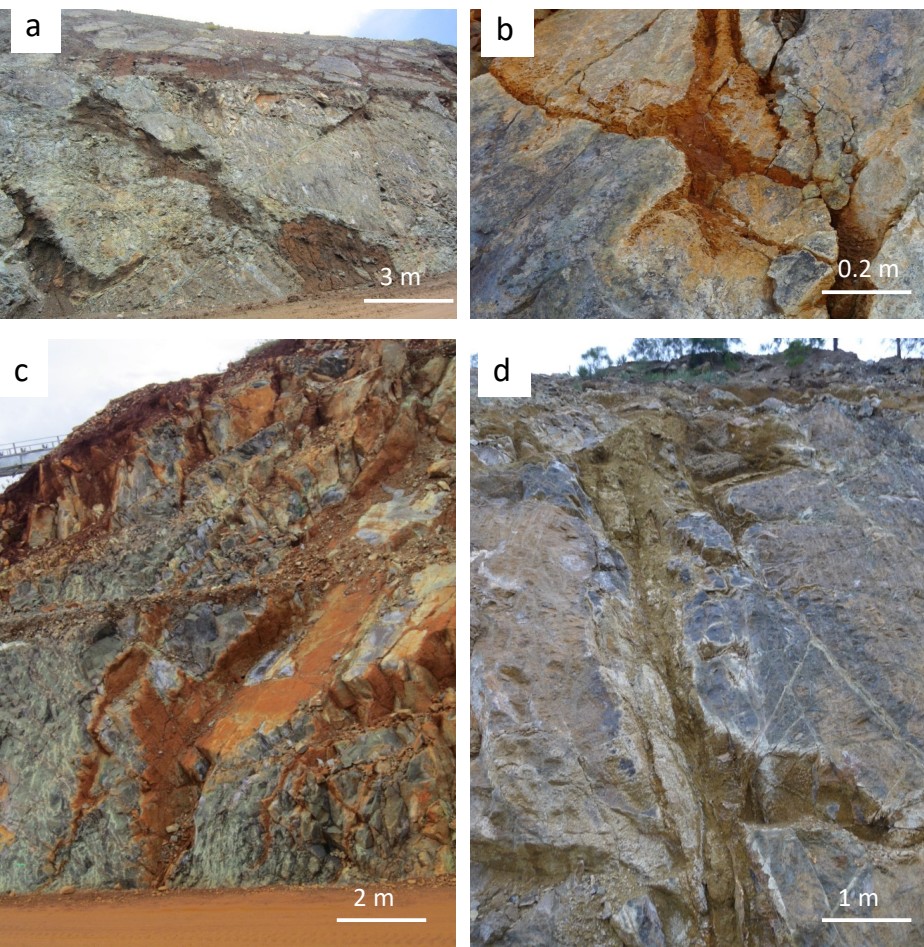

**Figure 2.** Dissolution pipes in the peridotite (main mining track at Koniambo). (**a**) Network of dissolution pipes near the high of the plateau. (**b**) Detail of the dissolution, which leaves only iron hydroxides as a coating on the edges of the pipe. (**c**) Vertical extension of the dissolution cavities (main track). (**d**) Dissolution along serpentinised fractures.

In the Cagou Pit, several dissolution pipes suffered complete collapse (Figure 3a,b). The cavities develop on the network of faults, the sides of which are usually serpentinised. They are filled with blocks of varied sizes. Blocks include weathered and silicified host rocks, serpentinised wall rocks, and fragments of kerolite crack-seals (Type I ores [24] which occur close to the serpentine fault. The breccias are cemented with white quartz and green pimelite (Ni-rich talc-like) (Figure 3c–e).

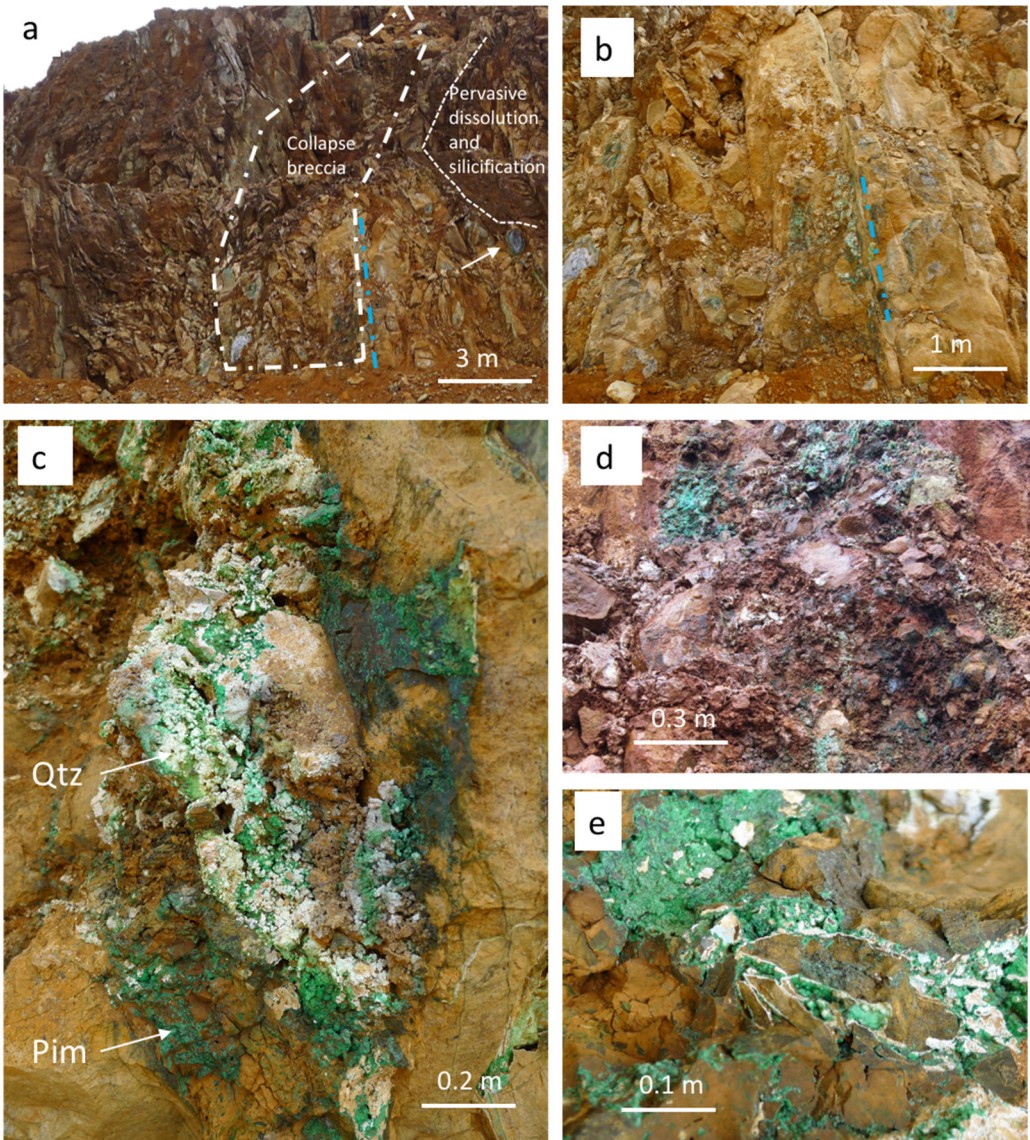

**Figure 3.** Collapse breccia (indicated by a white dashed line) in the open pit Cagou (former "207") from the Koniambo mining site. (**a,b**) Photographs were taken in the upper part of the open pit showing the penetration of the dissolution pipes, which are accompanied by pervasive silicification (dashed line) and quartz fracture infillings. The white arrow indicates a target like ore type II in (**a**). In (**a,b**), the blue dashed lines indicate fractures filled with blueish kerolite and red quartz (ore Type I), as described in [24]. (**c**) Cemented breccia by Ni-silicate (pimelite) and quartz. (**d,e**) Details of the cement. The arrows indicate the location of target-like ores (without quartz), as described in [23]. Qtz: quartz, Pim: pimelite.

### 4.3. Petrography and Mineralogy of Breccias

Collapse breccias constituted of blocks, including silicified host rocks and clasts of ore Type I, are cemented by quartz and pimelite (Figures 4 and 5).

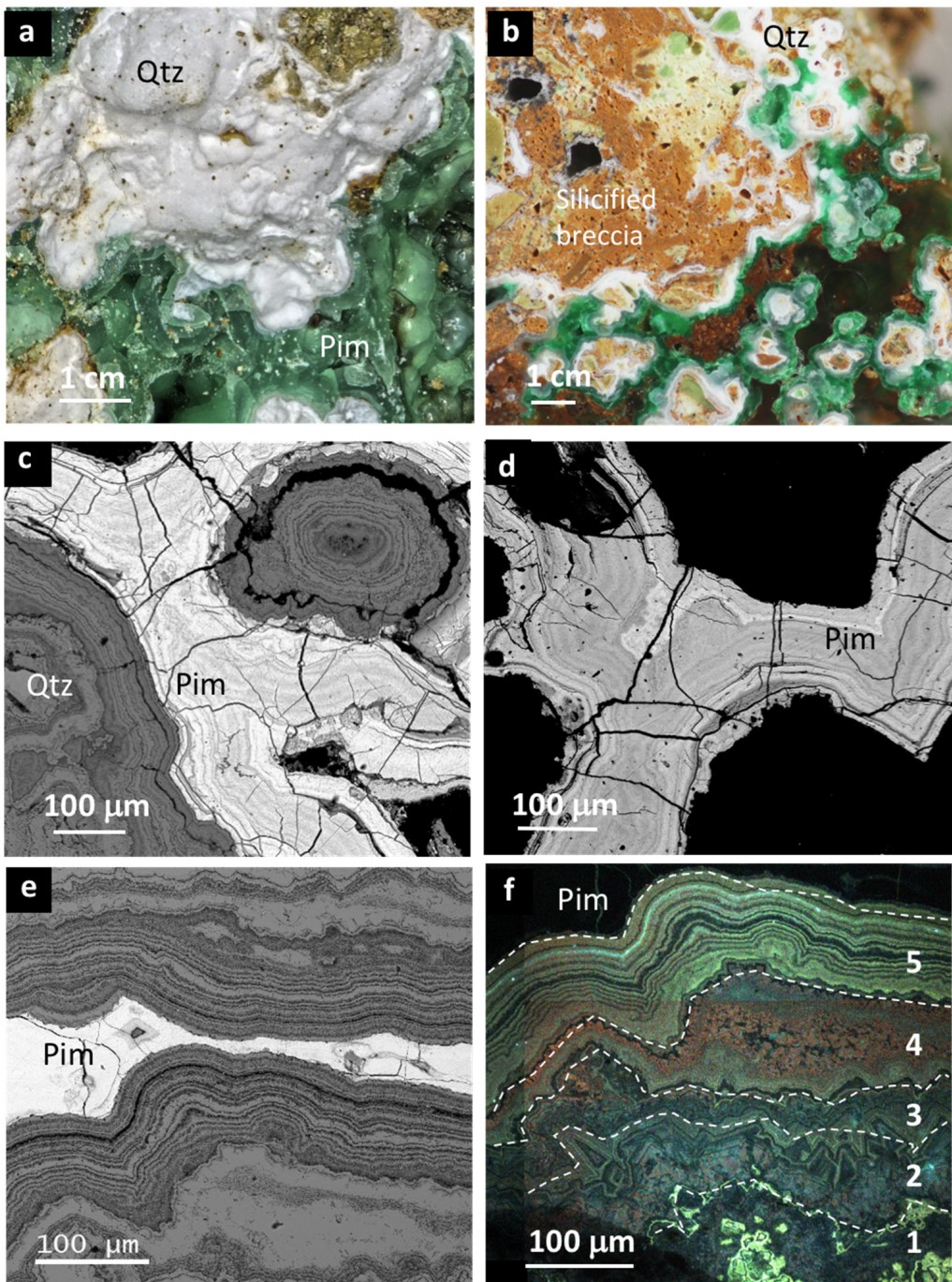

**Figure 4.** (**a**) Ni-silicate (pimelite, noted Pim) crystallised on white quartz (Qtz). (**b**) Pimelite on white quartz cementing silicified clasts. (**c**) BSE image showing the rims of quartz flowed by pimelite. (**d**) Detail of the chemical zoning in the pimelite colloform textures. (**e**) White quartz observed by BSE, showing a decreasing size of crystals; (**f**) Cathodoluminescence image showing white quartz rims from 1 (earliest rim) to 5 (colloform quartz).

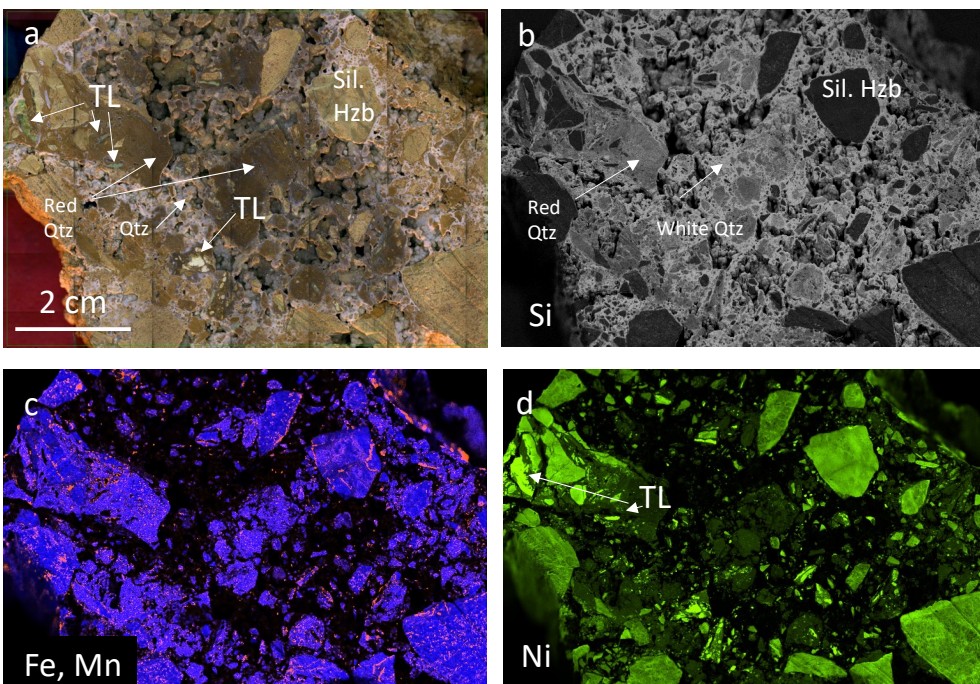

**Figure 5.** Micro-XRF images: (**a**) Silicified breccia sample from Cagou pit (Koniambo) with clasts of silicified harzburgite (Sil. Hzb), red-brown microcrystalline quartz (Red Qtz) and Ni-rich talc-like ore (TL). (**b**) Si map. (**c**) Superimposed Fe and Mn maps: Mn as Mn oxyhydroxides is located in microfractures and microvugs; Fe is present mainly in silicified harzburgite and red-brown microcrystalline quartz. (**d**) Ni map showing the location of mineralised clasts of ore Type 1 and clast of silicified harzburgite particularly enriched in Ni (TL: Ni-rich talc-like). All chemical images (**b**–**d**) are at the same scale than the macrophotograph shown in (**a**).

Breccia clasts include predominantly silicified rock fragments (noted silicified clasts) and a few clasts of ore Type I crack-seal infillings. The latter are brownish iron-rich microcrystalline quartz with relics of early blueish-green talc-like (Figure 4a,b). They are enriched in Ni and have micro-fissures and microvugs filled with Mn-oxides. Silicified clasts of host rocks are enriched in iron but are less rich in metals than red-brownish quartz. Micro-XRF images of a breccia sample (Figure 5a) and the corresponding chemical maps (Figure 5b–d) show that white quartz rims are almost free of metals, which are, on the contrary, relatively abundant in the other clasts. The clasts of microcrystalline red quartz are Fe-rich and contain spots and micro-fissures of Mn-oxides (Figure 5c). Most clasts, including silicified host-rock, contain Ni in their mass (Figure 5d), besides the visible ore clasts composed of Ni-Mg talc-like inclusions (noted TL for talc-like).

White quartz cement forms successive white, transparent quartz rims on the clasts and it is free from any other mineral inclusions. The late overgrowths are porous compared with the earlier quartz rims, as SEM back-scattered images show (Figure 4c,f). Cathodo-luminescence images also reveal thin layering with rims about 10 microns thick. Quartz luminescence intensity is relatively weak, with a 20 s exposure time required to obtain enough signals for imaging (Figure 4f). Quartz cementing micro-breccias close to the silicified clasts have light yellow luminescence. Brown-orange luminescence corresponds to an intermediate rim of subhedral quartz below the latest colloform rims. To the end of the quartz sequence, the most recurrent colour observed under cathodoluminescence is a greenish-blue colour for microcrystalline quartz alternating with quartz rims having a very low or absent luminescence appearing as black or dark grey.

Pimelite formed on the quartz rims and constituted the final filling of the breccia (Figure 3c,e and Figure 4d,e). This constantly develops over a succession of quartz cement increments described above, ending in a geode. Pimelite forms botryoidal layers of dark

green colour, showing strong contrasts of chemical mass (Z) in the section, displaying the growth zones (Figure 4c,d and Figure 6a). As shown in Figure 6a, magnesium is always lower than 2 wt% and has limited fluctuation, with NiO contents remaining between 43 and 49 wt%. Such Ni concentrations correspond to the extreme pole of the Ni-Mg kerolite solid solution (NiO = 39% to 49%) (Figure 6b). This pole is characterized by a high-frequency Raman spectrum marked by a principal OH band at 3650 cm$^{-1}$ and a shoulder at 3660 cm$^{-1}$, typical of pimelite [23]. As already observed in the case of target-like Ni-rich talc-like, pimelite does not show extended Ni-Mg substitution and is close to the end member of the solid solution.

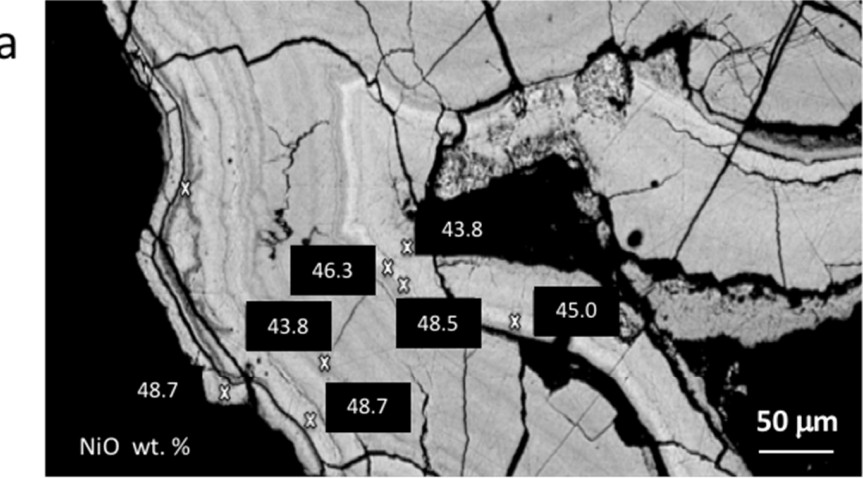

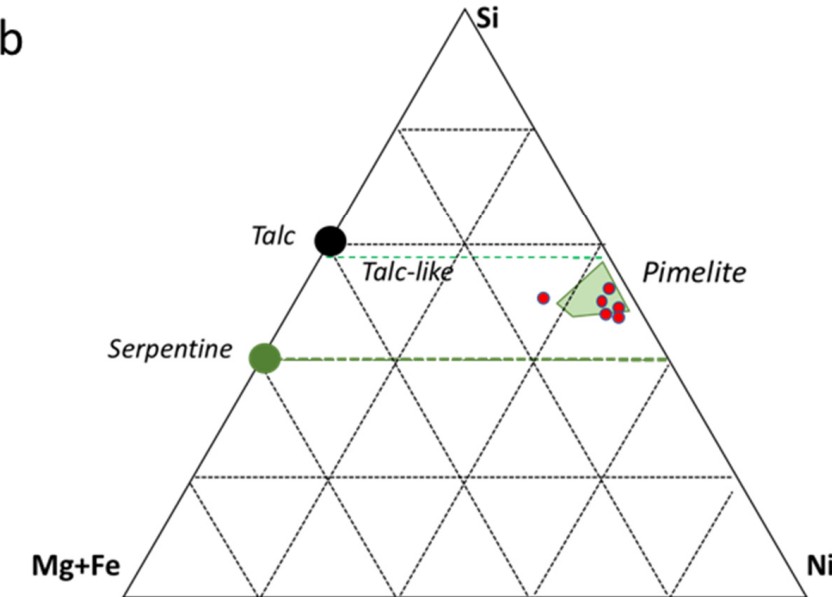

**Figure 6.** (**a**) Distribution of Ni (NiO%) in colloform pimelite. Each band has a high and relatively constant Ni concentration characterised by its specific mean Z (average atomic weight). (NiO concentrations from EMPA). (**b**) Si-Mg + Fe-Ni diagram applied to electron microprobe analyses of pimelites from Koniambo compared to pimelite from other New Caledonian sites (domain in green for other Caledonian deposits such as Thio and Poro, unpublished data). The reference line for talc-like solid-solution has already been proposed by [23,44].

When observed under TEM, the pimelite botryoidal texture consists of subparallel flexuous flakes (Figure 7a), which are sometimes associated with bi-pyramidal quartz (Figure 7b) but are primarily mono-mineral associations of tiny crystals of 50 to 150 nm in

length (Figure 7c,d). The HRTEM images show that the flakes are made of the piling of no more than fifteen to twenty layers, generally less [23,44]. The spacing is close to 10 ± 0.5 Å among incertitude due to particle orientation and typical of talc-like [23] (Figure 7e,f). Although well crystallised, the coherent domain is small, explaining the relatively smooth XRD patterns and the difficulties in obtaining good diffraction patterns under TEM. The pimelite flakes are not associated with serpentine-like layers, which are entirely lacking.

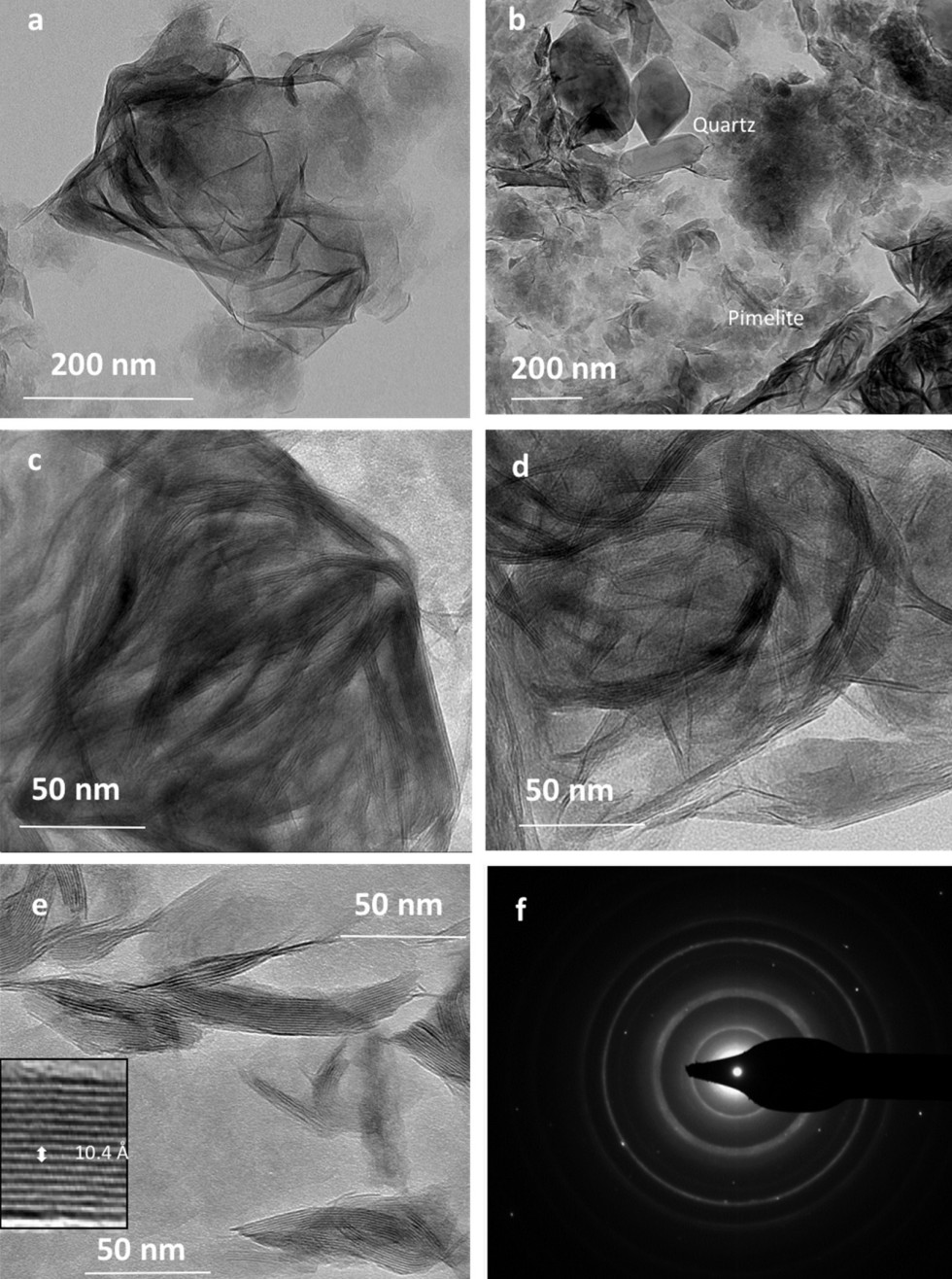

**Figure 7.** (**a**) TEM images of the Ni-silicate (pimelite). (**b**) Pimelite and euhedral quartz. (**c**,**d**) Flexuous assemblage of small size pimelite particles. (**e**) HRTEM image showing that the number of talc-like layers is generally low, about 5 to 15 in each particle, with no interstratification with other minerals. Indicative layer spacing is provided. (**f**) Diffraction pattern of one pimelite particle, with talc-like spacing.

### 4.4. Geochemical Data on Breccia Clast and Cement

Element concentrations in quartz types obtained by LA-ICP-MS were normalised to fresh surrounding rock (harzburgite) and presented in the box plots from Figure 8a. Red quartz has higher trace element concentrations than other quartz types, around one to two orders of magnitude higher. The richness in metals of the red microcrystalline quartz from the ore Type I is probably related to numerous metal-bearing micron-sized iron oxide particles. Al, Fe and Ga are two to five times higher in red microcrystalline quartz than in harzburgite. In white quartz, most elements are in low concentrations, around a few ppm, e.g., two to three orders of magnitude less concentrated than in the harzburgite, except Zn and Fe.

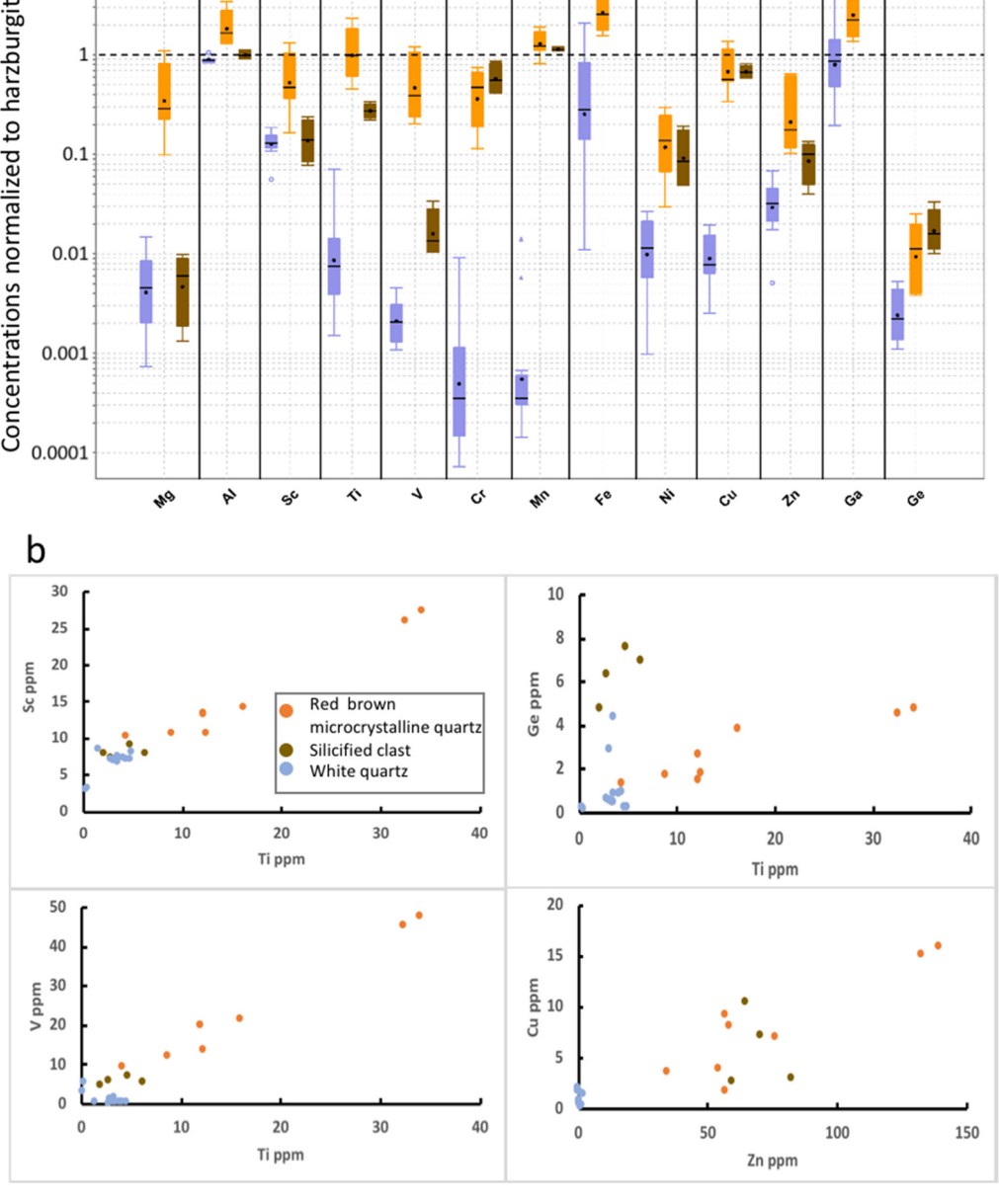

**Figure 8.** Trace elements in red-brown microcrystalline quartz, silicified clast and white quartz cement. (**a**) Normalised concentrations to harzburgite concentrations. (**b**) Binary plots, Sc-Ti, Ge-Ti, V-Ti and Cu-Zn for the three types of quartz.

The binary graphs, Sc-Ti, Ge-Ti, V-Ti and Cu-Zn established for the three types of quartz show that the red microcrystalline quartz has Ti, V, Cu and Sc concentrations of about two to three (up to ten) times those of the white quartz and are significantly enriched in Cu and Zn (Figure 8b). Ge concentrations are unusually high in silicified rocks, two to three times the concentrations determined in the other two types of quartz.

### 4.5. Quartz Oxygen Isotope Composition

Two $\delta^{18}O$ measurement profiles (twenty-five analyses) were made in the white quartz cement from the breccia. The $\delta^{18}O$ values of the white quartz range from +16.7‰ to +25.7‰ in the profile with twelve analyses shown in Figure 9. The second profile (13 analyses) was located near the same geode with $\delta^{18}O$ values ranging from +15.2‰ to +23.5‰. The $\delta^{18}O$ values evolve from +15.2‰ to +17‰ in the silicified breccia towards +24‰ to +25‰ in the centre of the geode, as shown by Figure 10a.

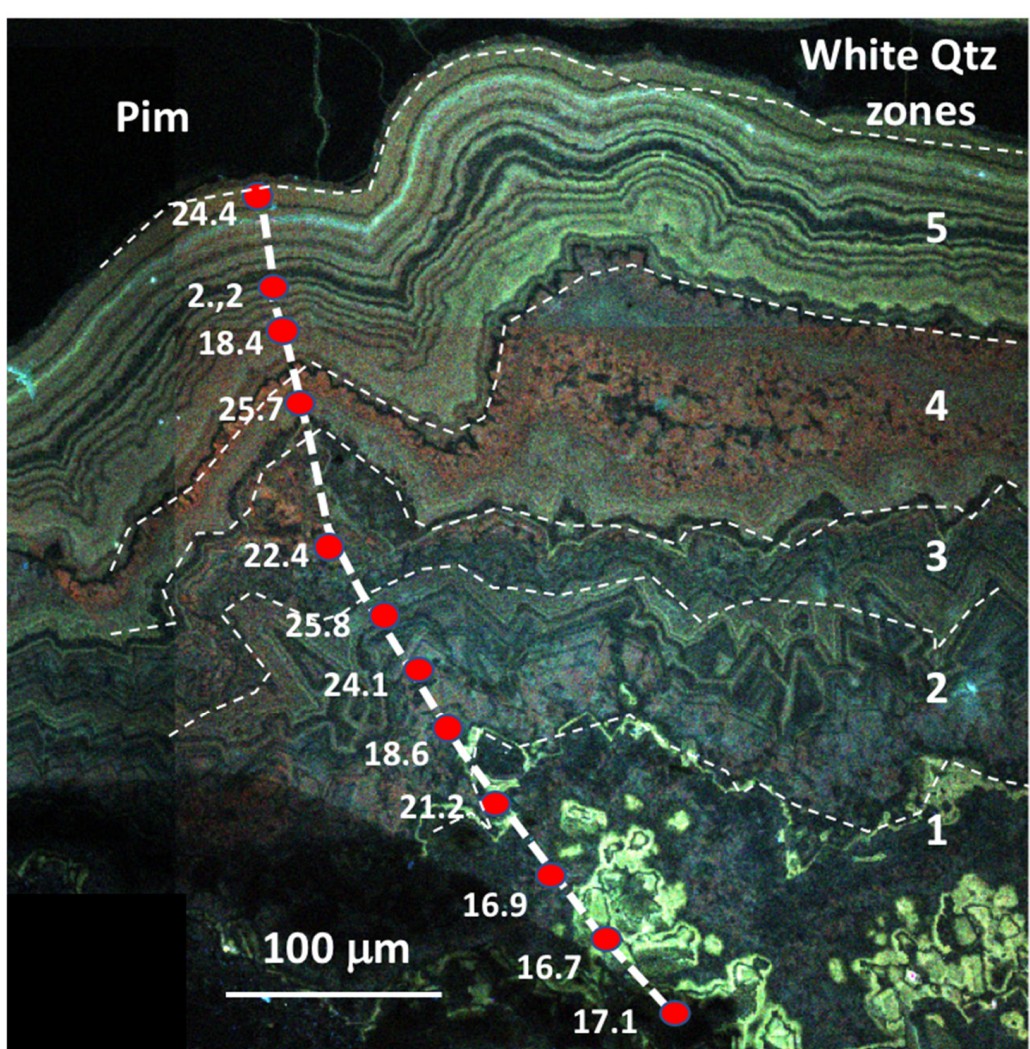

**Figure 9.** $\delta^{18}O$ (‰) values along a profile crosscutting white quartz from silicified breccia to the centre of the cavity (Zones 1 to 5). The colloform pimelite appears in black in the central vug. $\delta^{18}O$ values are reported on the cathodoluminescence image.

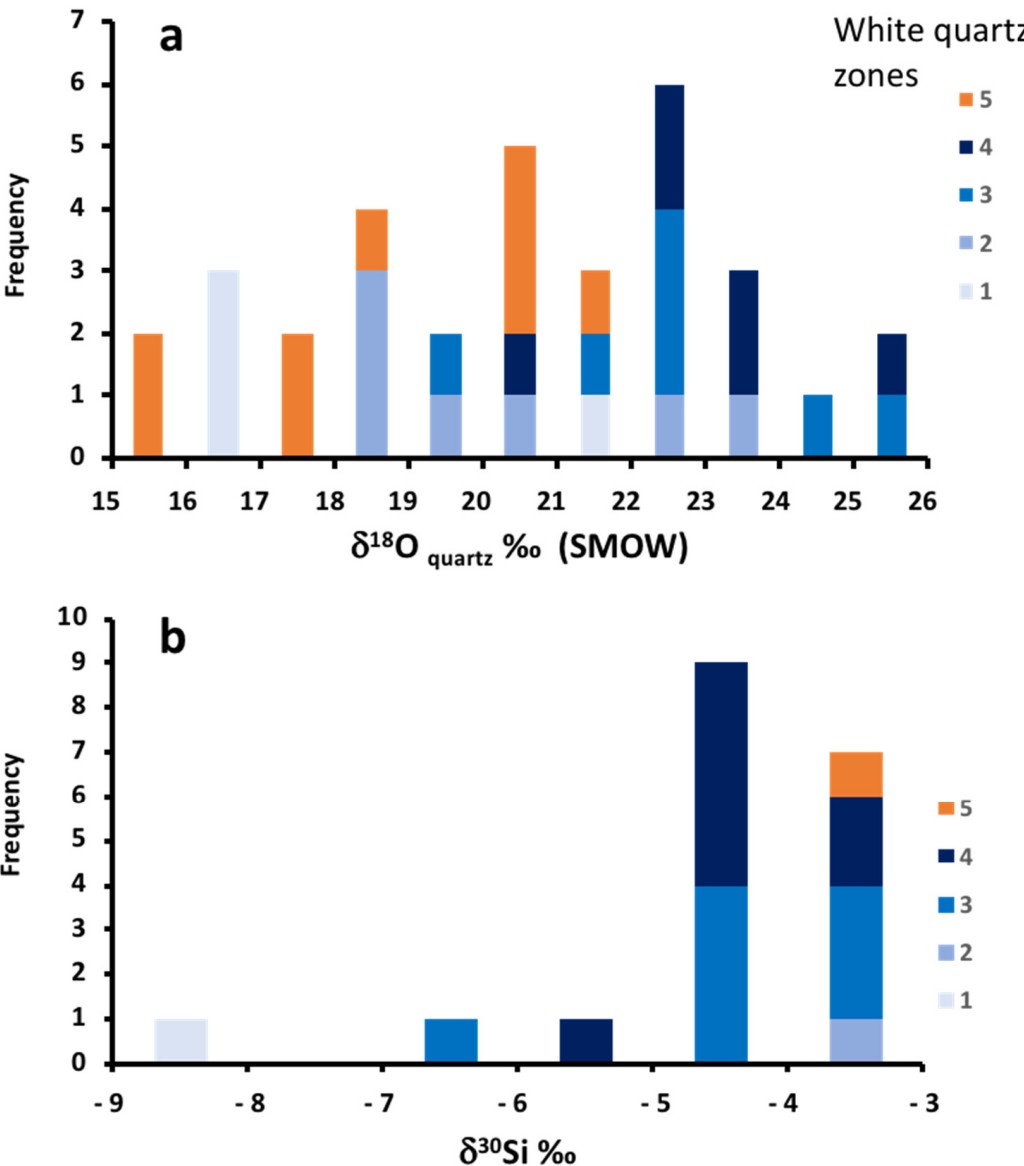

**Figure 10.** (**a**) $\delta^{18}O$ values along two profiles across the white quartz from the edge to the centre of the vug (five zones described in Figures 4f and 9). (**b**) $\delta^{30}Si$ quartz (‰) along the same profiles.

*4.6. Silicon Isotope Composition of White Quartz*

Analyses of $\delta^{30}Si$ were performed along the same profiles of white quartz (sixteen measurements, Figure 10b) as those used for the oxygen analysis spot (every two oxygen spots). The white quartz cement of the breccia sample shows a range of $\delta^{30}Si$ values from $-3.0‰$ to $-6.3‰$, with a mean value of $-4.8‰$ and a whole range of 3.3‰. A lower value of $-8.8‰$ characterises the earliest rim of white quartz.

## 5. Discussion
*5.1. Dissolution of Primary Silicates and Formation of Dissolution Pipes*

The dissolution pipes constitute an open network that may be considered a pseudo-karst. Such cavities in peridotite cannot result here from other processes than supergene dissolution, as shown by the mineralogy of the remaining iron (iron hydroxides) similar to that of laterite, and the dissolution features similar to subsurface boulders at the bedrock/saprolite boundary.

During intense rainfall, all primary silicates of the peridotites (olivine and pyroxene), the serpentines (early mesh and fracture infillings) and the secondary silicates related to the

saprolite (talc-like, nontronite) are dissolved. Part of the rainwater gullies down and part of it feeds the laterite water table with a gravity-driven flow in the slope direction, most often close to the interface between the laterite and the saprolite [16,30,31,45]. However, some water flows along sub-vertical to relatively steep faults through fracture sets in the damaged areas of major faults. The consequence is the expulsion of significant masses of silica, magnesium and nickel, preferentially in the transmissive fracture network. In some zones, silicification of the saprolite occurs, preferentially at the top of the relief. Such silicification of the weathered harzburgite is described in the Cagou pit [24,25]. Following Butt's (2014) discussion [46], silicified saprolites have been found at several locations, such as Cawse (Australia) and Caldag (Turkey).

Dissolution pipes develop where drained waters initialise the dissolution of primary and secondary silicates. Continued dissolution along fractures then leads to the formation of channels (pseudo-karstification of the peridotite). The preferential water movements yield an empty structure where around 10% of the initial rock forms a brown coating along the fracture, constituted of goethite. Such dissolution processes leaving open cavities are distinct from those associated with the subparallel deepening of weathering front in lateritic profiles where pores resulting from dissolution collapse progressively, yielding dense sub-horizontal laterite horizons at the basis of the weathering profile.

Along these drains, the interaction between the renewed rainwater and the fracture walls leads to a total dissolution of primary and secondary silicates, leaving only the insoluble iron as oxyhydroxide (goethite) along the walls. In the absence of collapse of the fracture edges, the drain widens until it reaches a width of several tens of centimetres, up to one metre locally. The downward deepening of the alteration depends on the gravitational force, the pre-existing fracture networks and the differential between the upper part of the profiles and the outlet zones in the valley. The channels thus form relays and allow the formation of a "karst" or "pseudo-karst" within the peridotite massif.

### 5.2. Chronology of the Karstification Related to River Incision

The river incision in the Taléa valley was studied, complementing earlier findings in the Koniambo and Kaféaté plateaux [20,42]. There was difficulty in distinguishing whether the relics in the Taléa valley belong to the S3 surface, the S4 surface or both. Their presence, nevertheless, suggests a strong relief inversion since S3 surface formation. The inversion was then enhanced during the S4 surface stage. The attribution to S3 and S4 can, nevertheless, be attempted based on the relative elevation between relics; the higher relics being S3 and the lower relics being S4.

If the paleo landforms of the Koniambo plateau are synchronous with that of Thiébaghi, an age of ca. 25 Ma for the S1-S2 surfaces weathering phase could be proposed. This would contrast with an early suggestion (ca. 30 Ma) by Chevillotte et al. (2006) [42]. Sevin et al. (2014) [47] considered, based on the sedimentary record in the Nepoui area, that a significant uplift event started around 22 Ma. Depending on location in New Caledonia, formations were uplifted of ~200 m at minimum to more than 500 m, explaining that the S1-S2 paleosurfaces occur at Koniambo around 800 m of altitude (>500 m). As a consequence, erosion started, and the valley incision quickly reached hundreds of meters. The beginning of the pseudo-karst development could date from that period. Subsequent valley incisions likely continued during the Neogene, leaving remnants of the S3 and S4 surfaces in the Taléa valley and their lower elevation equivalent in the Kaféaté plateau.

The breccia formation mechanism implies relief and a gap between the valley table elevation and the highs. After the first stages of laterite formation (S1-S2, most likely around 25–20 Ma), probably under more moderate relief, the effects of rock uplift brought these laterite profiles up to their current elevation (over 800 m). In this context, a series of erosional cycles occurred, leading to several stepped surfaces, as described by [19].

Pre-existing fracturing related to ante- to syn-obduction deformation stages was reactivated at several periods, notably during extensional stages from Miocene to the late Miocene. This extension is at the origin of the reactivation of normal faults of serpentine

planes, which are most often oriented in the island's axis, i.e., NNW–SSE, and generally a receptacle for fluids charged with Ni by the dissolution of both overlying rocks and pre-existing mineralisation. A tentative reconstruction of this dynamic evolution is proposed in Figure 11, with four corresponding stages: first, the late Paleogene period with two main plateau surfaces, followed during the Neogene period, by the formation of S3-S4 paleo-weathering surfaces synchronous of the karst formation and deepening.

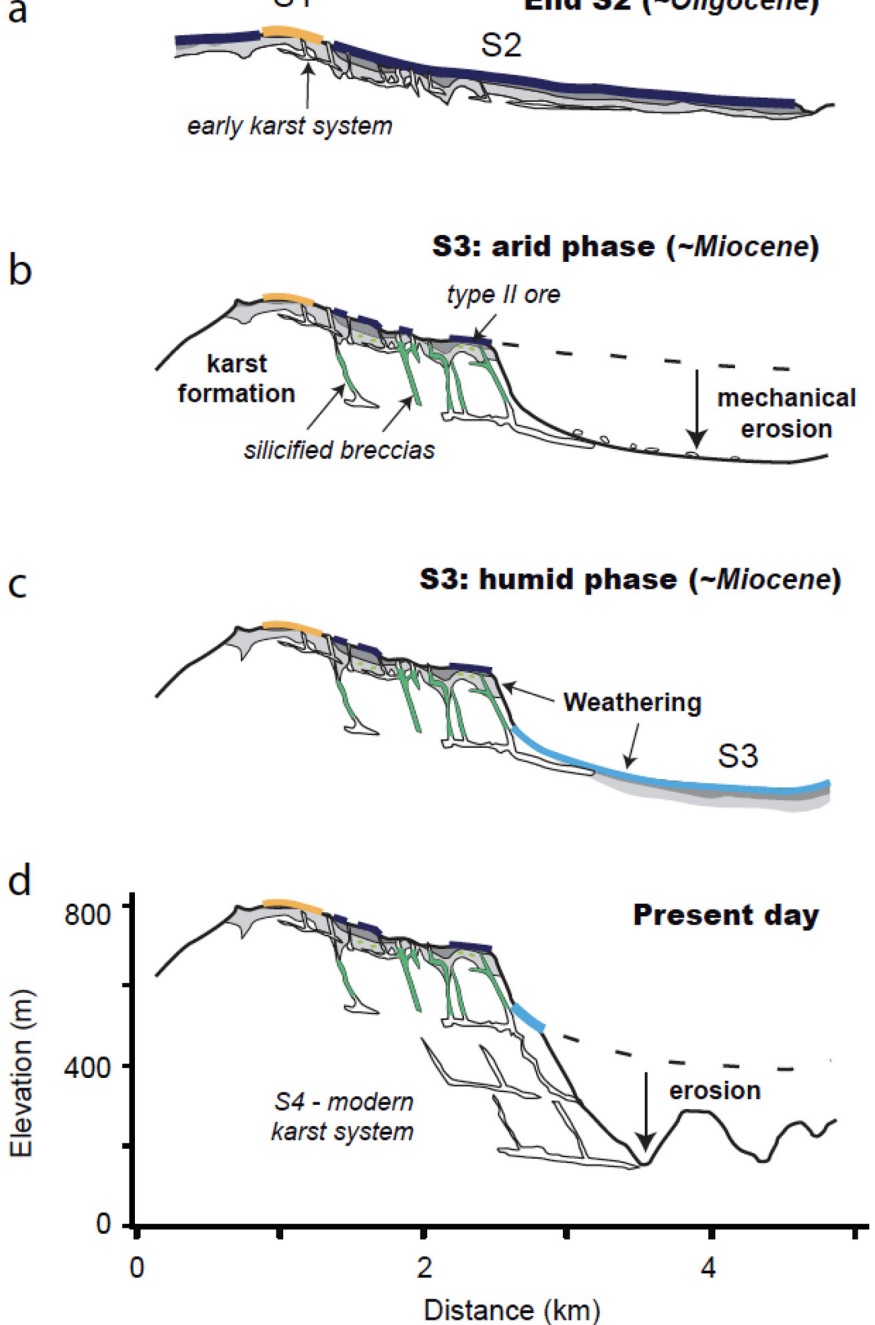

**Figure 11.** Conceptual model of the pseudo-karst development at Koniambo, with the paleosurface formation following the valley's deepening and progress in the erosion of the uplifted ophiolite.(**a**) Formation of the paleosurface S2 at the expense of S1, and initiation of the early karst system, (**b**) Deepening of the valley, and karst silicification during arid periods, (**c**) Formation of S3 related to humid periods, (**d**) Present day relief and karst.

The cavities result from the downward underdrawing, i.e., the gravity collapse, which is the principal driving force behind the brecciation. The resulting fill is a so-called collapse breccia, forming a "breccia pipe", as described in other geological contexts. The formation of these breccias is well constrained in the tectonic history, as they postdate the sealing of the kerolite crack-seals (Type I ores [24]) and are probably sub-synchronous with the formation of the target-like (Type II) ores [23]. They postdate the main laterite and are synchronous to the dissolution pipe formation, as they require a high dissolution rate along preferential meteoric water conduits. Thus, these breccias likely postdate S1-S2 surfaces, and are probably syn- to post-S3 at Koniambo.

### 5.3. Paleo-Climatic Conditions of the Formation of Quartz Cements

As quartz does not exhibit any studiable fluid inclusions but only tiny monophase fluid inclusions, generally considered metastable and formed below 60–80 °C, higher temperatures cannot be considered. The mineralogy of the pseudo-karst pipe is the same as around boulders issued from the supergene weathering of the peridotite in the saprolite, which produces only one run product in the laterite horizon: goethite. Goethite is not stable above 80 °C, as shown by thermodynamic data. If the activity of water is close to one, the boundary between goethite and hematite is around 35 °C, and the goethite-hematite conversion is favoured in the unsaturated zone explaining the formation of the hematite-rich duricrust on the laterite surface at the expense of goethite. Along the dissolution pipe, there is no evidence of hematite, but the only presence of goethite indicates that temperatures never reached high temperatures.

The relatively tight grouping of $\delta^{18}$O values for quartz suggests that the quartz formed from a fluid that has a similar oxygen isotopic composition during a single event. This event is distinct from the previous quartz stages characterized by different chemical features, particularly red microcrystalline quartz with hematite micro-inclusions and high metal contents. $\delta^{18}$O values between +18‰ and +24‰ would correspond to the lower values of the present-day waters around −4‰ within the 60–80 °C range. Such water values typical of tropical climate, with ambient temperatures of around $25 \pm 10$ °C, would yield $\delta^{18}$O values for quartz higher than +30‰, which were never found in New Caledonia. Only some opals were thus found with such values of +32–33‰ and correspond to recent silica films and draperies [25]. Either the temperatures were higher or the water $\delta^{18}$O was lower than the present-day ones. Temperatures around 70 °C were invoked for red microcrystalline quartz from crack-seals, supposed to be issued from low-temperature hydrothermalism, during tectonic compression events [24]. In the present case of very late cementing of quartz near the surface in pseudo-karst, a low temperature is the most realistic hypothesis. Ambient temperatures yield water oxygen isotopic values of −8 to −12‰, symptomatic of a cooler climate, as already invoked for Australian silcretes [48]. Adequate rainfall for the formation of the dissolution pipe is required. This could suggest a humid tropical or subtropical environment [27], but the alternation of seasons of rain and drying episodes could also happen under a less humid climate.

Considering that silicification was favoured by evaporation under ambient temperatures (mean temperature of $20 \pm 5$ °C), the $\delta^{18}$O values of the fluid are estimated to range between −12‰ and −8‰ (Figure 12). By comparison, $\delta^{18}$O values of chemical and possibly biogenic cherts range from +15‰ to +22‰, and $\delta^{18}$O of silicified volcanoclastic cherts between +9‰ to +17‰ [49]. Features of New Caledonian silicifications display similarities with silcretes. For instance, the silcretes from Australia have values of +24‰ to +29‰ [48,50]. In the case of Lake Eyre, late Eocene to early Oligocene silcretes are considered as precipitated at temperatures of 15–20 °C from waters having $\delta^{18}$O values of −6.9‰ to −12.2‰ [50]. Silcretes from Victoria and South Australia are thought to have precipitated from an undefined age between Eocene–Miocene to present at temperatures of 14–15 °C from rainwaters having $\delta^{18}$O values of −5‰ to −1‰ [48]. In other examples of silcretes, a value of $+29.3 \pm 1$‰ was found for quartz from silcretes in St. Peter sandstones (SW Wisconsin) which precipitated at 10–30 °C, from waters with $\delta^{18}$O of −5‰ to

$-10‰$ [51]. Similarly, Bustillo et al. 2014 [52] found $\delta^{18}O$ values of $+25$–$26‰$ for the silcrete found in Miocene sediments from Torrijos (Madrid basin, Spain), which are considered post-Miocene and precipitated at 15–25 °C, from meteoric waters with $\delta^{18}O$ of $-6‰$ to $-8‰$.

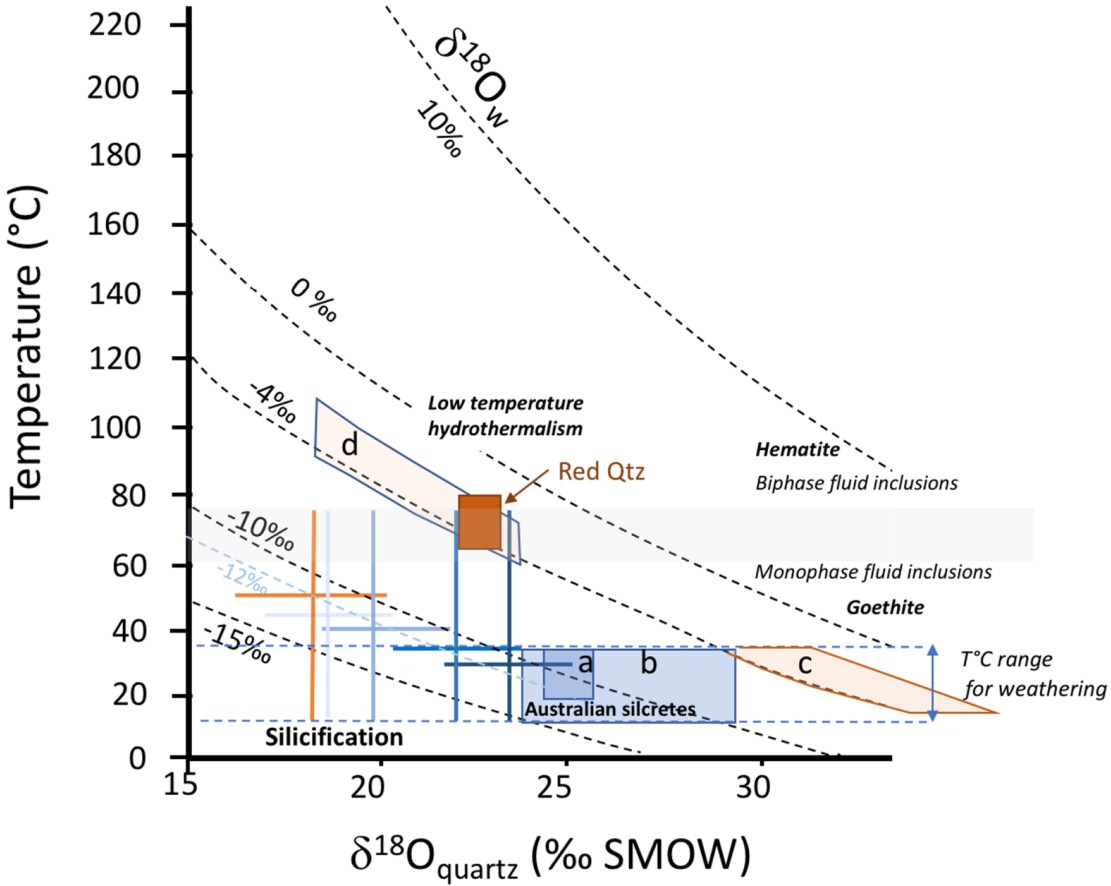

**Figure 12.** $\delta^{18}O$ quartz versus temperature diagram with reference lines for oxygen isotopic compositions of water ($\delta^{18}O$ w) reported from Matsuhisa et al. (1979) [53]. Data for the white quartz from Koniambo pseudo-karst are represented as crosses using mean values and ranges for the five zones described in Figures 4f and 9 with the same colours as in Figure 9. Temperatures are reported arbitrarily between 15 and 75 °C and crosses are organized to follow the water $\delta^{18}O$ value of $-12‰$ found in other silcretes. For comparison, $\delta^{18}O$ ranges and estimated temperatures for Lake Eyre silcretes are reported in the box a from [50] and other Australian silcretes in box b from [49]. Two domains calculated for present-day waters in tropical countries of $-2‰$ and $-4‰$ are reported for two temperature ranges (15–35 °C (domain c) and 60–110 °C (domain d). The estimated temperature range for red microcrystalline quartz is reported in the brown box. The boundary in grey indicates the limit of stability of goethite and hematite, as well the temperature fields for monophase and biphase fluid inclusions.

### 5.4. Analogies with Silcretes

The range of $\delta^{30}Si$ values for white quartz is quite similar to silcretes values, which range from $-6‰$ to $+1‰$ following [54]. This range is much lighter than most other Si reservoirs compiled in [55]. The number of available $\delta^{30}Si$-$\delta^{18}O$ is much more restricted, and most data concern archean cherts. The $\delta^{30}Si$ versus $\delta^{18}O$ diagram from Figure 13 confirms the specific features of silcretes and Koniambo pseudo-karst cements compared to most available silicon-oxygen isotope pairs, particularly those obtained on recent and old cherts and Icelandic hydrothermal systems [56]. A small number of silicon data, not plotted in Figure 13, overlap the range for silcrete and white quartz from Koniambo pseudo-

karst, principally the dispersed data over an extensive range of values obtained on quartz amygdules from Isua basalts [57]. However, the reason for the scattering of $\delta^{30}$Si is not fully understood, as the magmatic vesicles underwent hydrothermalism and metamorphism up to the amphibolite facies.

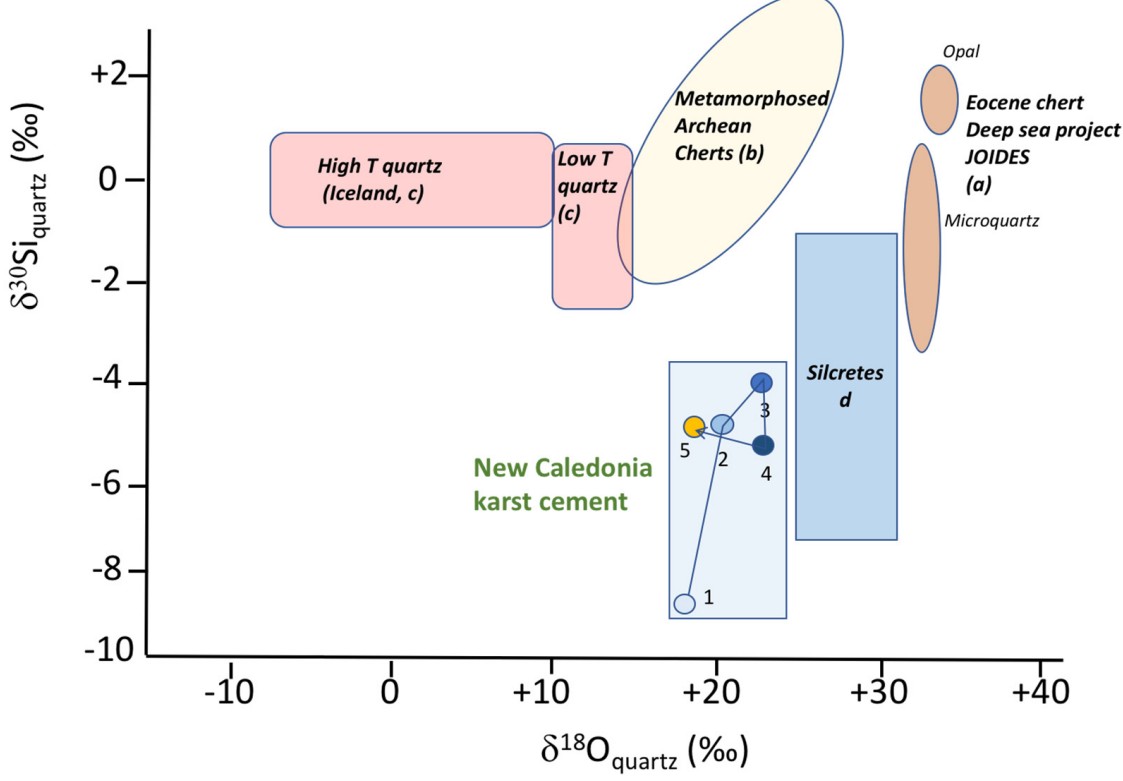

**Figure 13.** In situ $\delta^{30}$Si versus $\delta^{18}$O for white quartz from Koniambo silicified breccia (numbers refer to zoning from Figures 4f and 9), compared to the principal domains of values available for recent cherts (**a**—[49]) and metamorphosed Archean cherts (**b**—[58]), hydrothermal quartz of high and low temperature from Icelandic fields (**c**—[56]), and silcretes (**d**—composite box from several data sources [50,51,54]).

As the $\delta^{30}$Si values are similar to silcretes, quartz is most probably formed through evaporation, as decreasing temperature, the most straightforward way to deposit quartz, is difficult to invoke. The evaporation was probably favoured by the heterogeneous permeability of the breccia, characterised by large clasts and blocks yielding a relatively high bulk permeability at the scale of the pipe. Quartz cementation of pseudo-karst breccia pipes occurred thus probably during a climatic period different from the present-day conditions, as suggested by the oxygen isotope data. Silicification also affects microfracture rocks and quartz filling sets of microfractures at all scales with no specific orientation. The host rocks at Koniambo, on both sides of the breccia pipe, are highly silicified. In the upper levels of the quarry, the complete dissolution of the remaining silicates yields quartz boxworks made of hundreds of microfractures filled by quartz, the remaining parts of the rocks being entirely dissolved.

Although amorphous silica is considered the most probable silica form to precipitate at low temperatures, only quartz was found in the fissures and cemented breccia. The necessary transformation of amorphous silica into quartz, invoked by Thiry and Milnes [59], is a conceptual model that is not based on systematic observation of the transition between these two phases at the micrometer scale. Suppose opal or other amorphous silica phases are thermodynamically unstable and must transform by ageing to quartz; this does not explain why opal is found in some silcrete layers and quartz in other layers from the same

age in Australia. Precipitation of quartz instead of amorphous silica indicates mostly lower $H_4SiO_4$ activity in solution when it precipitates.

In addition, there are a lot of examples of quartz precipitation near the subsurface in the context of silicified caps in laterite and silicified peridotite, such as in United Arab Emirates [60], Turkey (Caldag laterite, [61]) and Australia (Mt Keith, [62]).

*5.5. Comparison with Silcretes from Australia*

In Australia, the timing of the silcrete formation is under debate. Some authors consider the main period of silicification as Eocene–Miocene, generally older than deep kaolinisation [26,63] and younger than the formation of ferruginous duricrust.

Van der Graaff (1983) [64] considers that Late Tertiary aridity cannot account for silcrete formation as silcretes predate Fe-duricrust. Considering North Queensland, Li and Vasconcelos [65] proposed an age of 14–10 Ma for a cooling period after the subtropical environment's 22 to 15 Ma period. More recently, Mathian et al. (2022) [66] obtained ages of 10 and 3.8 Ma to form kaolinite at Syerston. Such ages are close to the 10.8 Ma (and younger) data obtained by ESR on Queensland silcretes [67].

Seasonal climate or better alternate rain and drying episodes of very short durations may provide a favourable environment for silicate dissolution, followed by active silica (quartz) precipitation after lowering the water table level.

The data are, therefore, convergent about the formation of silcretes during the Miocene -likely after 15 Ma from waters having a $\delta^{18}O$ lighter than present ones under climatic conditions of moderate precipitation (<1 m/year) and average temperature. The climate was probably favourable both in Northern Australia and New Caledonia during this period to forming silcretes during the dry periods, alternating with rains under paleoclimatic conditions, which were different from the present.

*5.6. A Late Silicification Stage into the Ni–Ore Chronology*

The chronology of the development of the silicification found in the Koniambo massif includes several stages, summarized in Figure 14. During the first ones, dissolution affects the joint and fault network leaving boulders in the saprolite (Figure 14a) and then deepens along faults (Figure 14b,c). As pseudo-karst conduits widen, they can become progressively fragile due to the excessive size of the open pipes, and the collapse of the walls fills the conduits with blocks of any size and orientation (Figure 14d). The resulting collapse breccias are well timed in the tectonic history, as they postdate the sealing of the kerolite crack-seals (Type I) [24] and are probably sub-synchronous with the formation of the target-like ore type (Type II) [23].

The white quartz free of iron oxides attests to the total iron immobility during its crystallisation and the supergene feature of the process. The latter is thus very different from the formation mechanisms of the red quartz described in the earlier crack-seals, where iron was transported in the form of $Fe^{2+}$ by low-temperature hydrothermal fluids [24]. The silicification occurs after the earliest lateritic stage (S1-S2) as brownish clasts of laterites are cemented by quartz precipitated from dissolved silica (Figure 14d,e). Karstification is enhanced by relief and favoured by incision starting at stage S3. This requires the release of silica by chemical weathering, by the dissolution of the remaining silicates in the weathered horizons and along the pipes. The silica release is not related to developing sub-horizontal residual soil but to water-rock interactions in the pseudo-karst along the drainage zone (faults) and their micro-fractured, damaged zone. The lack of laterite contamination of the breccias indicates that the collapse process affects and dissolves rocks along the drains but is sufficiently far from sub-surface conditions. The karst, therefore, develops at depth, and the increasing incision of the valley favours the downward deepening of the pseudo-karst.

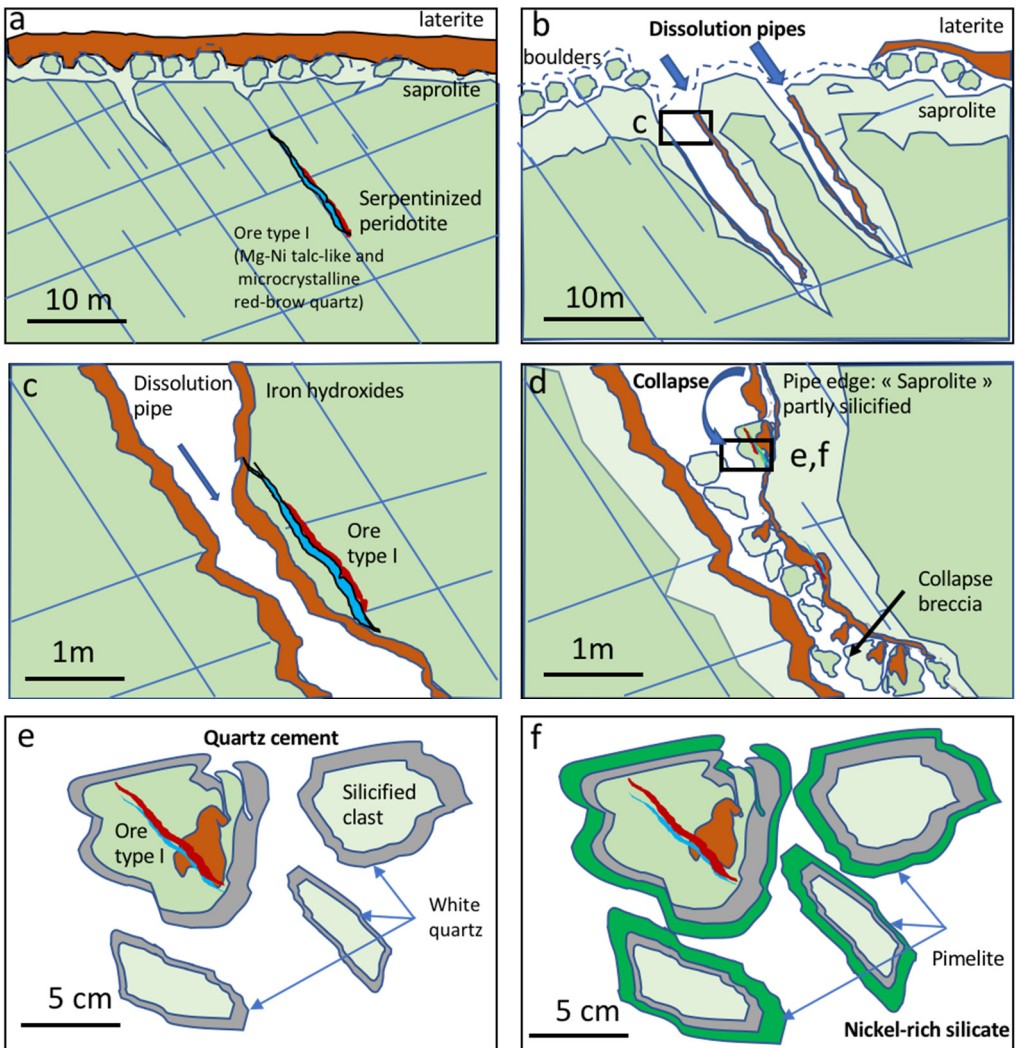

**Figure 14.** Main stages of silicified collapse breccia formation: (**a**) Formation of laterite and ore Type I as crack-seal fractures. (**b**) Development of dissolution pipes along previous fractures. (**c**) Detail of the dissolution pipes (inset c from b) with edges affected by saprolitisation and a layer of iron hydroxides. (**d**) Silicification and then the collapse of the dissolution pipes. The box refers to figures (**e**,**f**). (**e**) White quartz cement precipitation on clasts of silicified host rocks, and clasts of early ores. (**f**) Precipitation by evaporation of nickel-rich silicate (pimelite) on the white quartz rims.

White quartz formed onto the block surface is posterior to the quartz crack-seals found as brecciated clasts (Figure 14e). Correlations observed for red quartz are similar to those already determined in laterites, as iron hydroxides are enriched in all transition metals. The white quartz is almost free of metals. Variations in elements such as Ti, Ge, Al and V seem more dependent on the nature of the quartz and probably substituted to $Si^{4+}$ in the quartz lattice. The contrasted chemical features of the microcrystalline red quartz associated with ore Type I crack-seals compared to the white quartz and silicified clasts of host rocks indicate a drastic change in the conditions of quartz precipitation and mobility of metals. The metals mobilized together with iron, probably under reducing conditions during ore stage I [24], are no more trapped during the late white quartz stage. Pimelite occurs after quartz (Figure 14f) and the lack of recurrence indicates that the process occurred only once.

### 5.7. Geochemical Modelling of the Cement Formation

The model consists of the progressive 1D dissolution of a fully saturated 4.5 m long olivine column by rainwater, followed by a progressive evaporation step. Olivine composition was selected according to probe measurements on fresh olivine: $Mg_{1.865}Fe_{0.127}Ni_{0.008}SiO_4$

and its initial porosity of 1% was taken from [68]. The initial rainwater composition was taken from [2] (Table 1). The precipitation rate driving the advection was assumed to be 1500 mm/year, while the diffusivity was assumed to be spatially constant and equal to $4.16 \times 10^{-10}$ m$^2$/s in agreement with [16]. The points represented in Figure 15 consist of the chemical composition of the solution in the middle of the column. During the dissolution step, mass transport occurs along the cells of the column. Water-rock interactions are governed by olivine dissolution kinetics and thermodynamic equilibrium for forming newly-formed minerals [16].

**Table 1.** Initial solution composition for the evaporation step from [2].

| pH | Mg (mg/L) | Si (mg/L) | Ni (mg/L) | Fe (mg/L) |
|---|---|---|---|---|
| 7.34 | 1.15 | $5.30 \times 10^{-1}$ | $2.33 \times 10^{-3}$ | $5.04 \times 10^{-5}$ |

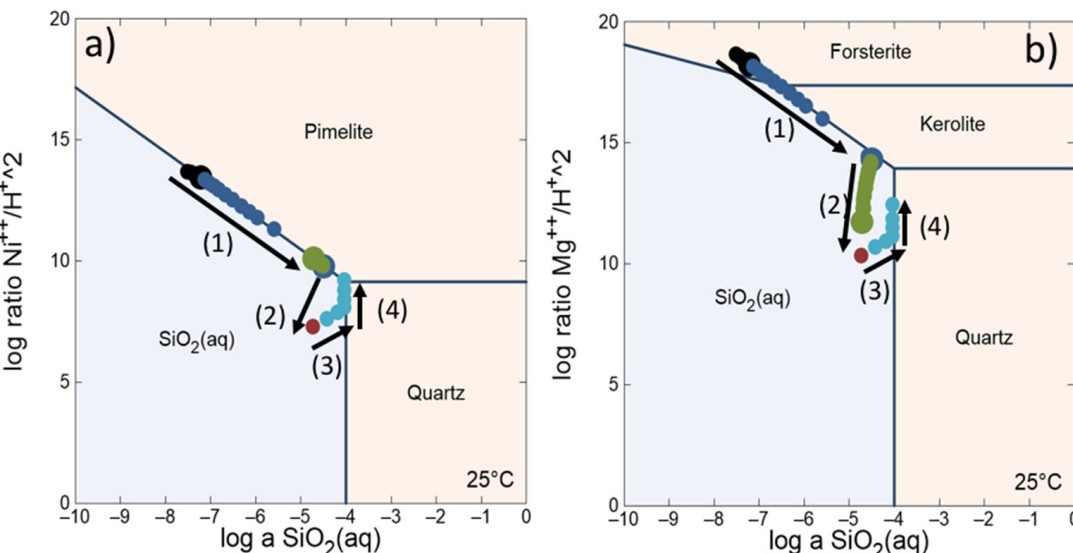

**Figure 15.** Chemical path of the mineralogical evolution in fractures in activity diagrams. (**a**) the Ni$^{2+}$/(H$^+$)$^2$ − SiO$_2$ and (**b**) Mg$^{2+}$/(H$^+$)$^2$ – SiO$_2$. In colours, the predominant species in the system: Black: Olivine; Blue: Mg-Kerolite; Green: Pimelite; Red: Goethite; Light Blue: Quartz. (1) Progressive silicate dissolution. (2) Goethite dissolution in the overlaying laterite. (3) Evaporation up to quartz saturation. (4) Evaporation up to pimelite saturation.

As the main driving force for silica precipitation (silcrete) at sub-constant ambient temperatures is evaporation [69], once the solution has reached equilibrium with goethite, another simulation is performed to model the progressive evaporation. For this step, the initial water composition is the solution in equilibrium with goethite. An arbitrary amount of water is removed from the previous solution at each stage. Then the saturation indexes of the considered minerals and the concentrations of Mg, Si, Ni and pH are calculated. These steps are repeated until only 0.2% of the initial water remains in the model to avoid changing the water activity from 1. The modelling has been carried out using either amorphous silica, in case of the formation of a precursor of quartz easier to precipitate at low temperature, or directly quartz. Results are not significantly different, except the inversion in the order of pimelite and silica-phase precipitation is not in agreement with field observation; therefore, justifying the choice of the quartz option in Figure 15.

After quartz precipitation, Ni silicate (pimelite) enrichment is explained by the redistribution of Ni from previous Ni silicates in fractures (ore stage 1) from the overlying levels and from harzburgite silicate dissolution. There is a need to increase the Ni$^{2+}$/(H$^+$)$^2$ ratio to reach oversaturation with respect to talc-like at sub-constant or slightly increasing silica activity in a solution (step (4) in Figure 15a). Notably, the solution reaches its saturation

with respect to the pimelite (Figure 15a) and not to kerolite for a partial evaporation state (Figure 15b). Hence, total evaporation seems to be never reached, explaining the absence of Mg-talc-like (kerolite) after the quartz precipitation, while pimelite is precipitated.

## 6. Conclusions

Breccias are posterior to the main laterite formation stage and are closely related to pseudo-karst development as they are collapse breccias requiring a high rate of preferential conduit dissolution. The preferential water movements along the fracture differ from the downward migration of the bedrock-saprolite development sub-parallelly to the bedrock surface. This corresponds to the progressive deepening of the pseudo-karst as a function of the valley incision with the following sequence:

The preferential subvertical drainage along pre-existing serpentine faults favoured the pseudo-karst network initiation. This episode corresponds to the formation of the S3 paleosurface. The pipe collapse formed breccias, which were then partly silicified and cemented by white quartz and pimelite, the extreme Ni-containing pole of the talc-like solid solution. The formation of this assemblage was favoured by a progressive increase of $Ni^{2+}/(H^+)^2$ and $Mg^{2+}/(H^+)^2$ ratios during interstitial water evaporation in the upper part of the pseudo-karst pipes. As shown by modelling, the fluid chemistry, however, reached only pimelite and not Mg-kerolite, explaining the predominance of pimelite over Mg-kerolite.

Quartz precipitation occurred in conditions identical to those invoked for silcretes. The low negative $\delta^{30}Si$ values ranging between $-5\permil$ and $-7\permil$ are typical of silcretes and close to the minimum values recorded worldwide. This silicification may have occurred during drier climatic episodes than in tropical climates from Eocene–Oligocene. Meteoric waters with lower $\delta^{18}O$ values in the order of $-6\permil$ to $-12\permil$ favour this hypothesis, which is proposed for the first time in New Caledonia.

The isotopic characteristics of the silicified breccias are similar to those of silcretes described in Australia, which are predominantly from the Miocene age. Ni reworking in the form of pimelite could have occurred at this period as New Caledonia was in a similar geographical and paleoclimatic position to north-eastern Australia.

**Author Contributions:** Conceptualisation, M.C. and M.-C.B.; methodology, M.C., M.-C.B. and J.-L.G.; software, J.-L.G. and S.F.; analysis, M.C. and M.-C.B.; writing—original draft preparation, M.C., M.-C.B. and J.-L.G.; review and editing, all authors; supervision, M.C., M.-C.B. and J.-L.G.; project administration, M.C. and Y.T. funding acquisition, M.C. and Y.T. All authors have read and agreed to the published version of the manuscript.

**Funding:** This work has been funded and logistically supported by the CNRT research contract CSF N° 9PS2017-CNRT. GEORESSOURCE/TRANSNUM« Facteurs d'enrichissements et transferts de Ni, Co-Sc dans les saprolites de Nouvelle Calédonie: approche géométrique, minéralo-géochimique et numérique », and by the French National Research Agency (ANR) through the national program "Investissements d'avenir" of the Labex Ressources 21 with the reference ANR-10-LABX-21-01/LABEX RESSOURCES21.

**Data Availability Statement:** Data will be available in the CNRT final report of the contract mentioned above, which will be in open access on the CNRT's website when completed.

**Acknowledgments:** The authors would like to thank Andreï Lecomte for cathodoluminescence imaging, J. Gambaja for TEM imaging (SCMEM, GeoRessources, Vandœuvre-lès-Nancy, France) as well as C. Peiffert (LA-ICP-MS facilities, GeoRessources), and Platform LG-SIMS at CRPG-Nancy in particular, N. Bouden and A. Gurenko for technical support. Sampling benefited from the help of Koniambo geologists, C. Couteau and participants to the first samplings (B. Quesnel, Ph. Boulvais from Géosciences Rennes). Three anonymous reviewers and the editor are warmly acknowledged for constructive comments.

**Conflicts of Interest:** The authors declare no conflict of interest.

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
