# Peer review of "Pseudo-Karst Silicification Related to Late Ni Reworking in New Caledonia"

_minerals, doi:10.3390/min13040518_

Round 1

Reviewer 1 Report

Rev minerals-2215116

The manuscript presents original results on siliceous formation linked to Ni ores in New Caledonia. The approach combines field work with nice petrographic and isotopic analysis. Although the results are novel, the manuscript contains many flaws.  The introduction does not address the issue and application for the mining industry and, more fundamentally, for the knowledge of the silica cycle.  The "result" section dealing with field work presents interpretation that should be displaced to the “discussion” section. Many concepts are not clearly defined and this does not facilitate the reading: dissolution pipes, breccia pipes; collapse breccias. The typology of quartz is confused. In the discussion section, many interpretations do not seem to be linked to the results. A major flaw concerns the comparison with silcretes which are not clearly defined here. The model of quartz precipitation at surficial conditions does not take into account the concepts currently assumed for silcrete formation. A conceptual model of the silicification and related Ni metallogeny should be given at the end.

1-2: “silicification ” should be removed in the title  because the process of silicification (via amorphous silica or not) is not addressed here (see comment later on)

29: a reference should be added at the end of the sentence 

34: same comment as above 

27-29: the importance of the issue should be addressed. In the abstract, comparison with Australian silcretes is done so the issue may be about the origin of quartz in laterites and the link with Ni ore deposits.

33: same as 29

41-45: the hypothesis to be tested or the addressed questions should be presented here.

55-56: the statement is not clear: before what? 

56: remains of what? The sentence should be rewritten.

74 : “dissolution pipe”  should be defined and a reference added

77-79: is it the main question addressed in the paper? If yes, parts of the M&M section may be displaced in the Introduction section

77: “breccia cement” should be defined; is it similar to “clasts….” described at 74-75

90: the authors should explain what they mean by “several ways”

148-178: observations only should be mentioned in the “results” section and interpretation displaced to a discussion section 

180: see at 74

182-190: see at 74 in the M&M section; is the concept of “dissolution pipe” from the present authors or from others? This point should be clarified because here many interpretations are given which should not be the case in a “result “section but rather in a “discussion” section

196-206: “breccia pipes” is a concept not defined before; what is the difference with “dissolution pipes”? Here again, observations and interpretations should be separated.

204-206: the mineral description should be moved to the next section (4.3)

216-217: “collapse breccias” should have been defined before; are they breccias filling “dissolution pipes” or “breccia pipes”?

226: “newly formed quartz cement”:  to be replaced by “newly-formed euhedral quartz”; the authors should explain why it is “newly”? 

236: “quartz cement”: it is similar to previously described “euhedral quartz ribbons”;  “white quartz”; “quartz rim”? the quartz typology should be defined.

23: which types?

246: the “quartz sequence” is not defined before; this should be clarified, indeed it is stated “to the end of the sequence….for microcrystalline quartz…” while  at 220-221 it may be suggested that microcrystalline quartz are older that euhedral type: to be clarified (typology and chronology).

258: “quartz rim”: is it a new type or similar to “euhedral quartz ribbons”? to be clarified

275: is “spheroidal texture” similar or different from “botryodal layers “at 260? to be clarified?

276: the “bi-pyramidal texture “should be indicated in figure 7a using an arrow, for instance.

296 :“ white microcrystalline quartz” is not described before: to be clarified (see problem of quartz typology already mentioned above)

298: the three types of quartz should be clearly defined (see problem of quartz typology already mentioned above); 3 types are given in the legend of figure 8 but there are not linked to the text: what about white quartz? euhedral quartz ribbons? quartz rim? white microcrystalline quartz? The location of the samples should be given: in space (figure 1b) and along the profiles (figure 2 and 3). 

311: the authors should use the same terminology everywhere: here “micro quartz“ is used: what is it if we refer to the 3 types used before? 

328: here “silicified clast” is used but what is it? Not in the types given in figure 8; this should be clarified

332: again, evidence for a “newly-formed” white quartz should be given 

332-33s4: there are no isotopes in Chardon and Chevillotte (2006) and Chevillotte et al. (2006); to be corrected.

336-338: may contradict the previous sentences; the authors should present the similarities between their data and the silcretes described in the literature (see at 333); 

341-343: based on the present study, what are the evidences that lead to this interpretation?

347: based on the present study, what are the evidences that lead to this interpretation?

(see also at 226, 332)

354-357: based on the present study what are the evidences that lead to this interpretation?

359-375: here again, there is no link with the data presented in this study: the evidences that lead to this scenario should be given; at 371 “alteration” should be replaced by “weathering”; the title of this section does not match the text: the mobility and deposition of Si is not discussed; to be modified

376-378: is it also possible to imagine other scenarii? for instance low temperature hydrothermalism (Monin et al., 2014, Biogeoscience) as a source of Si?  

380-381: already mentioned at 51-52; to be removed.

381-382: what are the evidences for 2 plateaux? This statement is not mentioned before; to be clarified.

380-405: based on the present study what are the evidences that lead to this interpretation?

410: “silcrete”: according to the discussion at section 5.1, it was not clear if the present siliceous formations are silcretes (see comment at 336-338); the authors should  clarify and give the  definition of silcrete 

429-430: do the authors mean “compatible with many examples of silcretes”? if “many others” is used, what are they?

448-449: “complete dissolution of the remaining silicates yields quartz boxwork…”: if the silicates were totally dissolved, what explains the boxworks? Do the authors mean that silicate dissolution and quartz precipitation are contemporaneous? To be clarified

458: the authors do not provide with mass balance calculation along the profil, so what are the evidence for “total iron immobility” and the “supergene feature of the processes”; what are the evidences to reject hydrothermal process?

464: here “silica” is used; do the authors mean that quartz is not the only silica phase? If not, quartz should be used

470-478: in silcretes, it is assumed that quartz formed through ageing of amorphous silica; for instance, the authors may refer to Thiry and Milnes (2016) Journal of Archaeological Science

490-492: a reference is needed

499-514: I agree that the “main driving force for silica precipitation at low temperature is evaporation”  but it leads to the formation of amorphous silica (opal A) not quartz; the literature about silcrete formation including the processes of silicification at low temperature should be discussed; here is what Thiry and Milnes (2016) say: “The most soluble phases in a geochemical system control the precipitation of the less soluble ones and, in this way, amorphous silica in equilibrium with a solution is able to sustain the crystallization of every other crystalline silica phase”; accordingly, Figure 14 is not adapted to silcrete formation and should be removed. Instead, I suggest a conceptual model of silicification with Ni redistribution. 

Author Response

see attached document

Reviewer 2 Report

This manuscript presents an original study on the source of silicification and dissolution associated with Ni-rich silcrete soils in New Caledonia.

Altough being of interest and sufficiently clear, the paper presents some major problems that are addressed in the pdf attached in this report with comments.

The main problems rise with the use of the terminology "karst", the interpretation and classification of the dissolution features, and the interpretation and presentation of the isotope values that require more attention. Also, the analytical design for the isotopic study should be improved to add significant data and discussion to be able to constrain the genesis of the different silica phases that were described.

Reviewer 3 Report

The submitted manuscript “minerals-2215116”, Cathelineau et al., is an interesting paper dealing with silcrete formation in karst. However, major revision is required prior the paper can be considered for publication.

In detail:

-         line 36: “In the Koniambo Massif, New Caledonia,” instead of “In the Koniambo Massif,”

-         line 50: the statement “Laterites consist of iron oxyhydroxides, iron being the only non-mobile element” is based on old fashioned references, I suggest introducing this issue in a greater detail and, possibly, including updated references.

-         lines 68-69: is the statement “The mobility of iron and associated metals characterizes this stage.” consistent with the one in line 50?

-         Lines 270-274: please add references for the pimelite from other New Caledonian sites.

-         Lines 292-294: there is something wrong or missed in the statement “Al, Sc, Co, V, and Ti concentrations normalised to the concentrations in the fresh surrounding rock (harzburgite) are two to five times higher in red microcrystalline quartz than those of harzburgite (Figure 8a).”. Do the Authors refer to the median? The maximum value? The third quartile? Further it seems to me that Sc and V are not, overall, particularly enriched with respect to the harzburgite whereas  no box plots for Co are available.

Again, what about the enrichments of Fe, Mn, Cu, and Ga?

This section is dramatically poor and it needs to be re-written and further detailed.

-         Sections 5.1 and 5.2 may be merged and condensed.

-         The discussion in section “5.5. Comparison with silcretes from Australia” is mostly based on the activity diagrams reported in Fig. 14. Do water data for the diagram’s implementation come from Butt, 1985? If this is the case it would be useful to better detail the context of the analyzed water. I guess the activity diagrams were made by the Geochemists’ Workbench software or a similar code, this issue must be reported in the Materials and Method section.

-         Lines 530-533: “The formation of this assemblage was favoured by a progressive increase of Ni2+ / (H+)2 and Mg2+/(H+ ratios during interstitial water evaporation in the upper part of the karst pipes. The fluid chemistry, however, reached only pimelite and not Mg-kerolite, explaining the predominance of pimelite over Mg-kerolite.”.

In this statement must be clearly indicated that water data come from previous research in a different geographic context.

Round 2

Reviewer 1 Report

The manuscript has been greatly improved. There are still some minor revisions that could be taken into account:

- the manuscript should be revised for  English language editing; for instance, I have noticed many inappropriate uses of "the" before nouns ( ex at 258, 259, 260 : "the pimelite"; 263: "the magnesium"...)

-202-207: this section contains interpretations that should be moved to the discussion section

-294: "the concentrations of the" should be removed

-397, Fig. 11: the ages of the 4 stages should be indicated

-258, 402, 466, 534: "deposited"  should be replaced by "formed" 

-to my previous comment at 499-514 concerning the formation of quartz at 25°C, the authors answered "Although amorphous silica is considered to be the most probable silica form to precipitate at low temperature, there are a lot of surficial examples of quartz precipitation". I suggest to add references.

Author Response

Thank you for your suggestions. see attached file

Reviewer 2 Report

The revised version of the paper answered to some of the problems pointed in the previous manuscript. However, i still do not believe in the interpretation of the authors of the isotopic values, which are not supported by independent geological data, but are relatively ideologic and speculative. The possibility of higher temperature to form the quartz can't be ruled out just stating that there are no biphase fluid inclusions. I believe a more robust discussion is still needed. By the way, i think that after major modifications (see attached pdf for puntual comments) the paper could be published.

Moreover, the paper is still lacking in a clear petrographic description of the different ore types (type-I vs type-II for istance). Some part of the discussion need to be improved and are difficult to follow. Also, some figure requires modifications. Finally, i reccomend a slight variation of the paper's title.

All the best,

Author Response

Thank you for your suggestions

see attached file

Reviewer 3 Report

The Authors improved the ms according to the suggestions and the paper

can be accepted as it stands.

Author Response

Thank you for previous comments and approval.